# Six-photon upconverted excitation energy lock-in for ultraviolet-C enhancement

Qianqian Su [1,6✉], Han-Lin Wei[1,6], Yachong Liu[1], Chaohao Chen [2], Ming Guan[3], Shuai Wang[1], Yan Su[4], Haifang Wang[1✉], Zhigang Chen[5] & Dayong Jin [2,3✉]

Photon upconversion of near-infrared (NIR) irradiation into ultraviolet-C (UVC) emission offers many exciting opportunities for drug release in deep tissues, photodynamic therapy, solid-state lasing, energy storage, and photocatalysis. However, NIR-to-UVC upconversion remains a daunting challenge due to low quantum efficiency. Here, we report an unusual six-photon upconversion process in $Gd^{3+}/Tm^{3+}$-codoped nanoparticles following a heterogeneous core-multishell architecture. This design efficiently suppresses energy consumption induced by interior energy traps, maximizes cascade sensitizations of the NIR excitation, and promotes upconverted UVC emission from high-lying excited states. We realized the intense six-photon-upconverted UV emissions at 253 nm under 808 nm excitation. This work provides insight into mechanistic understanding of the upconversion process within the heterogeneous architecture, while offering exciting opportunities for developing nanoscale UVC emitters that can be remotely controlled through deep tissues upon NIR illumination.

[1] Institute of Nanochemistry and Nanobiology, Shanghai University, Shanghai, People's Republic of China. [2] Institute for Biomedical Materials & Devices (IBMD), Faculty of Science, University of Technology Sydney, Sydney, NSW, Australia. [3] UTS-SUStech Joint Research Centre for Biomedical Materials & Devices, Department of Biomedical Engineering, Southern University of Science and Technology, Shenzhen, Guangdong 518055, People's Republic of China. [4] Genome Institute of Singapore, Agency of Science Technology and Research, Singapore City, Singapore. [5] State Key Laboratory for Modification of Chemical Fibers and Polymer Materials, College of Materials Science and Engineering, Donghua University, Shanghai, People's Republic of China. [6] These authors contributed equally: Qianqian Su, Han-Lin Wei. ✉email: chmsqq@shu.edu.cn; hwang@shu.edu.cn; dayong.jin@uts.edu.au

Multiphoton upconversion processes that convert NIR excitation into visible emissions have attracted considerable attention owing to broad technical applications of anti-Stokes shifts[1–6]. UV upconversion luminescence can be a powerful tool for applications in biomedical[7–9], environmental[10,11], and industrial fields[12,13], and converting NIR all the way upto UVC (100–290 nm) emissions holds promise in photocatalysis[11], ultraviolet solid-state lasers[12], and biomedical applications[8,14–17]. But, their practical implementations have been hindered by low emission intensities and difficulties in achieving large shifts into the UVC region. Apart from the intrinsic parity-forbidden nature of 4f−4f optical transitions in lanthanide systems, NIR-to-UVC upconversion can be significantly influenced by many deleterious factors, such as concentration quenching, surface quenching, cross-relaxation between lanthanide ions, and competitive energy harvesting from lower-lying energy levels. To minimize the unwanted energy consumption at high-lying emitting levels and reduce the chances for mitigating the upconverted UV emissions, attempts have been made to enhance the emission intensity in the UV range, for instance, by controlling the particle phase and size[18], the pulse width of excitation beams[19], dopant composition[20], and nanoparticle core-shell structures[12,21–24]. To our best knowledge, little attention has been paid to the effect of interior defects on UVC upconversion luminescence[25].

Compared with $Yb^{3+}$-sensitized upconversion nanoparticles (UCNPs), $Nd^{3+}$-sensitized UCNPs offer deep penetration depths and minimal over-heating effect, owing to low coefficients of water absorption under 800-nm excitation[26]. $Nd^{3+}$-sensitized UCNPs are the promising candidates for photon-driven reactions in biosystems, such as biodetection[27], photodynamic therapy[28–31], light-triggered drug release[32], and photocatalysis[33]. To enhance the brightness of $Nd^{3+}$-sensitized UCNPs, core-shell nanostructural design has been typically utilized to prevent deleterious cross-relaxation[34–37]. By doping lanthanide ions and $Nd^{3+}$ ions into the separated layer, the emission intensity can be notably enhanced while maintaining optical integrity[38]. Despite enticing prospects, UVC emission from $Nd^{3+}$-sensitized UCNPs has been challenging because of the densely packed excited states of $Nd^{3+}$ and dominant cross-relaxation within the nanoscale systems[39].

Here we report the significantly enhanced UVC emission through $Nd^{3+}$ sensitization by controlling upconverted excitation energy flux within $Gd^{3+}/Tm^{3+}$ codoped core and multishell nanostructures. Our mechanistic investigation reveals an upconverted excitation lock-in (UCEL) mode in which $Gd^{3+}$-sensitized excitation energy can be retained by simply using an interlayer of the $NaYF_4$ host lattice doped with $Yb^{3+}$ that is optically inert to the excited $Gd^{3+}$. This nanostructure preserves the upconverted UV energy within the core domain and effectively suppresses energy dissipation by interior traps, enabling six-photon-upconverted UV emission at 253 nm under 808 nm excitation.

## Results

**Heterogeneous nanostructural design.** In our experiment, we designed a heterogeneous core-multishell structure to suppress surface quenching and achieve tunable emissions. In a conventional design[35,40], under 808-nm excitation, $Nd^{3+}$ sensitizers harvest excitation photons and subsequently pass them to $Yb^{3+}$ ions with an excited state at ~10 000 cm$^{-1}$. Energy migration through a network of high concentration $Yb^{3+}$ ions promotes energy transfer of the NIR excitation to $Tm^{3+}$ emitters with ladder-like metastable intermediate states, facilitating sequential upconversion processes from NIR to visible/UV. Subsequently, upconverted UV emission from high-lying states of $Tm^{3+}$ can be

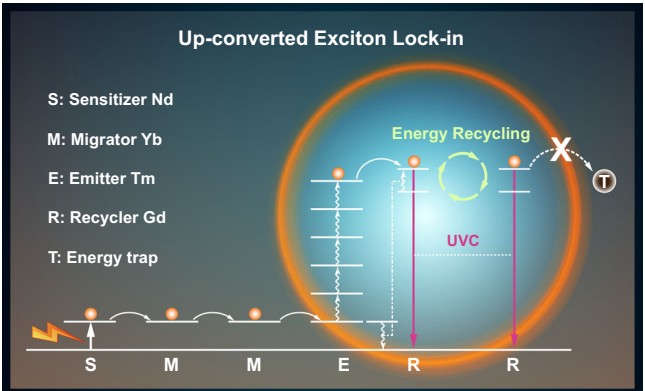

**Fig. 1 Schematic illustration of upconverted excitation lock-in (UCEL) mechanism for UVC generation within a nanoparticle.** The proposed UCEL scheme involving a heterogeneous, core-multishell nanostructure (Gd-$CS_YS_2S_3$). A multistep cascade energy transfer ($Nd^{3+} \rightarrow Yb^{3+} \rightarrow Tm^{3+} \rightarrow Gd^{3+}$) leads to populate the excited states of $Gd^{3+}$. The layer of an optical inert $NaYF_4$ host lattice doped with 20% $Yb^{3+}$ locating in the first shell layer of nanoparticles can lock-in the upconverted excitation energy of $Gd^{3+}$ ions and prevent depopulation by deleterious energy traps within the nanoparticles, resulting in intense UVC upconversion emission. S, M, E, R, and T denote sensitizer $Nd^{3+}$, migrator $Yb^{3+}$, emitter $Tm^{3+}$, recycler $Gd^{3+}$, and energy traps, respectively.

further transferred to $Gd^{3+}$ ions embedded in the nanoparticle core as the UVC energy reservoirs.

The key to our design is the use of a $NaYF_4$ host lattice doped with the same amount of $Yb^{3+}$ locating in the first shell layer of $NaGdF_4$:49%Yb, 1%Tm@$NaGdF_4$:20%Yb@$NaGdF_4$:10%Yb, 50% Nd@$NaGdF_4$ (Gd-$CS_{Gd}S_2S_3$) nanoparticle (Fig. 1). This layer of $NaYF_4$:20%Yb is optically inert to the excited states ($^6D_J$, $^6I_J$, and $^6P_J$) of $Gd^{3+}$ ions and can lock-in the upconverted UVC and ultraviolet-B (UVB) energy of $Gd^{3+}$ ions. The $Gd^{3+}$ network can then reuse the upconverted excitation energy and prevent depopulation by deleterious energy traps within the nanoparticles, as well as absorb additional photon energy from the excited state $Yb^{3+}$ ions. The $NaYF_4$ layer plays a key role in interdicting detrimental energy transfer between $Gd^{3+}$ and interior traps, enhancing five- and six-photon-upconverted UVB and UVC emissions.

**Upconverted excitation lock-in (UCEL) mode.** The UCEL mode requires both an interlayer of optical inert $NaYF_4$ host lattice doped with $Yb^{3+}$ and a network of $Gd^{3+}$ ions to recycle upconversion energy for UVC emission amplification. Fig. 2 illustrates a typical upconversion process in the heterogeneous core-multishell nanoparticles upon 808-nm excitation. The 808 nm photons are first sensitized by $Nd^{3+}$ sensitizer ions, being populated at the $^4F_{5/2}$ energy state and quickly relaxed to the $^4F_{3/2}$ energy state of $Nd^{3+}$. The excited $Yb^{3+}$ ions serve as an energy migrator to sensitize and pass on the energy from $Nd^{3+}$ and to populate the $^3P_2$ state of $Tm^{3+}$ through a five-photon upconversion process. Subsequently, the energy at the $^3P_2$ state, relax non-radiatively to populate $^1I_6$ and give rise to UVB emissions at 290 nm. Besides, $Gd^{3+}$ ions in the core domain extract the energy through an energy transfer process of $^1I_6 \rightarrow ^3H_6$ ($Tm^{3+}$): $^8S_{7/2} \rightarrow ^6P_J$ ($Gd^{3+}$). The excitation energy of $Gd^{3+}$ at $^6P_J$ can resist nonradiative quenching due to its large energy gap (~32 000 cm$^{-1}$ from $^6P_J$ to $^8S_{7/2}$). Thus, the lifetime of $Gd^{3+}$ at $^6P_J$ energy state is long enough for the sixth photon to be absorbed from the excited $Yb^{3+}$. Therefore, the $^6D_J$ state of $Gd^{3+}$ is further populated by the appropriate energy matching of the following transitions of $^2F_{5/2} \rightarrow ^2F_{7/2}$ (9750 cm$^{-1}$, $Yb^{3+}$): $^6P_J \rightarrow ^6D_J$

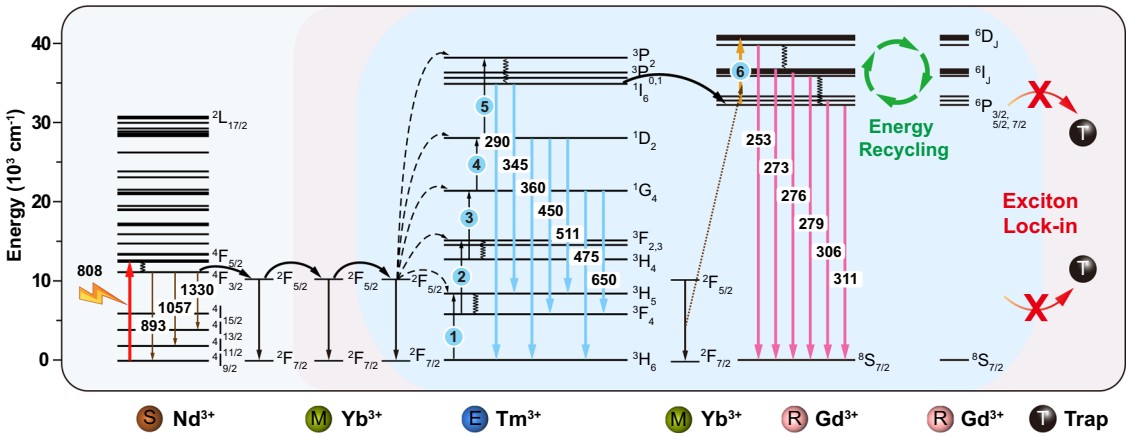

**Fig. 2 Schematic energy diagram of heterogeneously doped lanthanide ions and their cascade energy transfer within a core-multishell nanoparticle.**
When the nanoparticles are excited under 808 nm, $Nd^{3+}$ sensitizers first absorb the excitation energy and pass it onto $Yb^{3+}$. Subsequently, the $^3P_2$ state of $Tm^{3+}$ is populated by a sequential five-photon energy transfer from the network of excited $Yb^{3+}$ ions and relaxes to $^1I_6$. The $^6D_J$ state of $Gd^{3+}$ is populated via a stepwise process of a five-photon energy transfer process from $Tm^{3+}$ and a further energy transfer from $Yb^{3+}$, giving rise to the sixth-photon upconversion luminescence. The inert $NaYF_4$ host lattice layer can lock-in the $Gd^{3+}$ excitation energy and reuse the energy that would otherwise be depopulated by deleterious energy traps within the nanoparticles, resulting in upconversion emissions in the UVB and UVC regions.

($\sim 8750\ cm^{-1}$, $Gd^{3+}$)[41–43]. Thus, UVC and UVB upconversion emission peaked at 253, 273, 276, 279, 306, and 311 nm from $^6D_J$, $^6I_J$, and $^6P_J$ of $Gd^{3+}$ can be obtained. Noted that, the probability of nonradiative relaxation of $^6D_J$, $^6I_J \rightarrow {}^6P_J$ is larger than that of the radiative transition of $^6D_J$, $^6I_J \rightarrow {}^8S_{7/2}$, resulting in an efficient population of the $^6P_{7/2}$ state, commonly observed in Gd-based homogeneous nanostructures[22]. In our design, the $NaYF_4$-based first shell layer selectively blocks the energy transfer from $Gd^{3+}$ to interior energy traps (e.g., lattice defects and impurities). It preserves and recycles the excitation energy within the core region, leading to increased populations in the $^6D_J$, $^6I_J$, and $^6P_J$ states of $Gd^{3+}$ and intense UVC and UVB emissions of $Gd^{3+}$.

**Controlled synthesis**. We used a layer-by-layer epitaxial growth method[24] to synthesize a batch of $Gd\text{-}CS_YS_2S_3$ nanoparticles with optimized concentrations of co-dopants[40] following the design of $NaGdF_4$:49%Yb,1%Tm@$NaYF_4$:20%Yb@$NaGdF_4$:10%Yb,50% Nd@$NaGdF_4$ (Fig. 3a). Transmission electron microscopy (TEM) images of obtained $Gd\text{-}CS_YS_2S_3$ nanoparticles show the average size of $\sim$29 nm with each layer $\sim$2.5 nm in thickness (Supplementary Fig. 1). High-resolution TEM shows the single-crystalline structure of the as-synthesized core-multishell nanoparticles (Fig. 3b inset), and X-ray powder diffraction result (XRD, JCPDS file number 27-0699, Supplementary Fig. 2) confirms the hexagonal phase of the as-prepared nanoparticles. High-angle annular darkfield scanning TEM identified the formation of the heterogeneous core-multishell structures (Fig. 3b), in which the brighter regions correspond to heavier elements (Gd, Yb, and Nd) and the darker parts correspond to lighter ones (Y). Energy-dispersive X-ray mapping analysis further confirms the heterogeneous core-multishell structures (Fig. 3c and Supplementary Fig. 3).

**Remarkable UVC enhancement**. To investigate the unusual UVC upconversion emission from $Gd^{3+}$, we recorded the photoluminescence spectra of the as-synthesized nanoparticles at room temperature. Usually, in favor of the lower $^6P_{7/2}$ (311 nm) energy level, the $Gd^{3+}$ emission in the UVC range is quenched, and optical transitions of ($^6D_J$, $^6I_J$, $^6P_{5/2} \rightarrow {}^8S_{7/2}$) could hardly be spectroscopically detected (Supplementary Figs. 4–7)[12]. In contrast, as shown in Fig. 3d and Supplementary Fig. 8, intense upconversion emissions from $^6D_J$ and $^6I_J$ of $Gd^{3+}$ peaked at 253 nm ($^6D_{9/2} \rightarrow {}^8S_{7/2}$), 273 nm ($^6I_J \rightarrow {}^8S_{7/2}$), 276 nm ($^6I_J \rightarrow {}^8S_{7/2}$),

279 nm ($^6I_J \rightarrow {}^8S_{7/2}$), 306 nm ($^6P_{5/2} \rightarrow {}^8S_{7/2}$) and 311 nm ($^6P_{7/2} \rightarrow {}^8S_{7/2}$) in the UV region were observed either under 808 nm or 980 nm excitation. Moreover, we observed more than 50-fold and 30-fold enhancements in $Gd^{3+}$ emission (311 nm) by our $Gd\text{-}CS_YS_2S_3$ heterogeneous core-multishell design compared with the conventional $Gd\text{-}CS_{Gd}S_2S_3$ nanoparticles under 808 and 980 nm excitation, respectively (Supplementary Figs. 9 and 10), although the absorption profile of $Gd\text{-}CS_YS_2S_3$ is not changed compared with that of $Gd\text{-}CS_{Gd}S_2S_3$ nanoparticles (Supplementary Fig. 11). The approximate absorption cross-section $\sigma$ of $Nd^{3+}$ at 808 nm was calculated to be $\sigma = 1.5 \times 10^{-19}\ cm^2$ ($Gd\text{-}CS_YS_2S_3$), $\sigma = 1.3 \times 10^{-19}\ cm^2$ ($Gd\text{-}CS_{Gd}S_2S_3$) from the UV−Vis absorption spectra of the nanoparticles[44]. As verified by the emission spectra of as-prepared nanoparticles from different batches of (Supplementary Fig. 12), our protocol to enhance the UVC upconversion emissions is reproducible.

We further studied the excitation power dependence of luminescence intensity from higher-lying $^6D_J$, $^6I_J$ and $^6P_J$ excited states of $Gd^{3+}$ (Fig. 3f). The number of photons ($n$) required to populate the upper emitting state can be calculated by the luminescence intensity $I_f$, and the pump power of laser $P$ following the relation of $I_f \propto P^n$[45]. The output slope for 253 nm emission band was calculated as 6.29, indicating that six 808 nm photons were needed to populate the $^6D_J$ level, following a six photon upconversion process (Fig. 3g), while $n$ values obtained for 276 and 311 nm emissions were 5.27 and 4.94, indicating five-photon processes (Supplementary Fig. 13).

**Quantitative study**. The large energy gap of about $32\,000\ cm^{-1}$ of $Gd^{3+}$ and intrinsic low phonon energy of $NaGdF_4$ offer good possibilities to obtain 100% energy transfer efficiency from $Gd^{3+}$-to-$Gd^{3+}$[46,47]. The energy transfer efficiencies $\eta$ of $Nd^{3+}$-to-$Yb^{3+}$, $Yb^{3+}$-to-$Tm^{3+}$, and $Tm^{3+}$-to-$Gd^{3+}$ energy transfer can be quantitatively estimated from the Eqs. 1 and 2[48,49]

$$\eta = 1 - \frac{\tau_m}{\tau_{Ln}} \tag{1}$$

$$\tau_m = \frac{\sum \alpha_i \tau_i^2}{\sum \alpha_i \tau_i} \tag{2}$$

where $\tau_m$ is the mean lifetime of energy donor lanthanides (Ln) in the presence of energy acceptor, $\tau_{Ln}$ is the intrinsic lifetime of energy

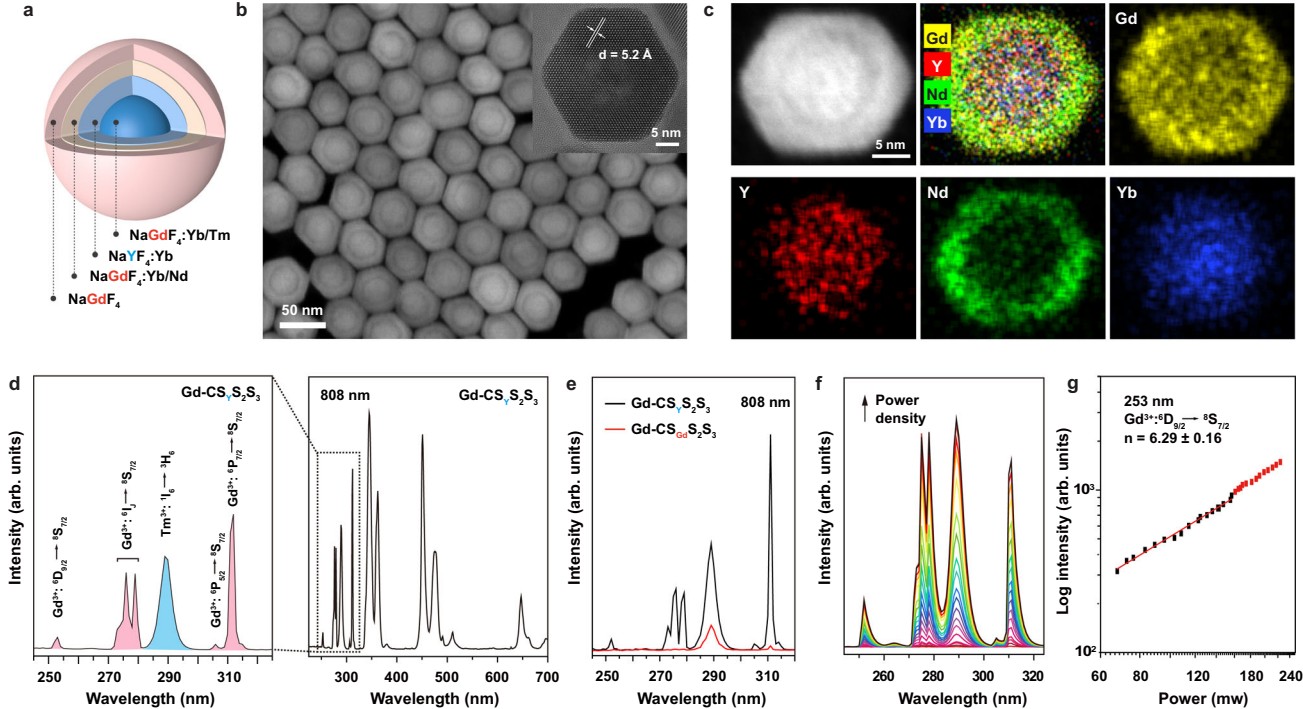

**Fig. 3 Structural and optical characterizations of Gd-CS$_Y$S$_2$S$_3$ nanoparticles before and after cation exchange. a** Schematic illustration of the as-synthesized Gd-CS$_Y$S$_2$S$_3$ nanoparticles. **b** High-angle annular dark-field scanning transmission electron microscopy (HAADF-STEM) image and high-resolution TEM image of the corresponding nanoparticles (inset). **c** HAADF-STEM image and elemental mapping of a single Gd-CS$_Y$S$_2$S$_3$ nanoparticle, indicating the spatial distribution of the Gd, Y, Nd, and Yb elements in the core-multishell structure. **d** Room-temperature emission spectra of Gd-CS$_Y$S$_2$S$_3$ nanoparticles in cyclohexane under 808 nm excitation. **e** Emission spectra of Gd-CS$_Y$S$_2$S$_3$ and Gd-CS$_{Gd}$S$_2$S$_3$ under excitation of 808 nm CW diode laser. The excitation power density is 10 W cm$^{-2}$. **f** Excitation-power-dependent UV upconversion emission spectra of Gd-C$_Y$S$_2$S$_3$ nanoparticles under 808 nm excitation. **g** Log intensity-pump power of the 253 nm upconversion emission of Gd-C$_Y$S$_2$S$_3$ nanoparticles under 808 nm excitation.

donor, and $\alpha$ is the amplitude. To calculate the energy transfer efficiencies of Nd$^{3+}$-to-Yb$^{3+}$, Yb$^{3+}$-to-Tm$^{3+}$, and Tm$^{3+}$-to-Gd$^{3+}$, we designed and synthesized three pairs of heterogeneous nanoparticles (TEM results shown in Supplementary Fig. 14). In our experiment, to first determine the intrinsic lifetime of the corresponding energy donors, the energy acceptors of Yb$^{3+}$, Tm$^{3+}$, and Gd$^{3+}$ were replaced by optically inert Y$^{3+}$ ions.

In detail, to calculate the energy transfer efficiency of Nd$^{3+}$-to-Yb$^{3+}$, we produced a pair of samples of NaGdF$_4$:49%Yb,1%Tm@NaYF$_4$:20%Yb@NaGdF$_4$:10%Yb,50%Nd@NaGdF$_4$ (Gd-CS$_Y$SS in the presence of 20% Yb$^{3+}$ energy acceptor) v.s. NaGdF$_4$:49%Y,1%Tm@NaYF$_4$@NaGdF$_4$:10%Y,50%Nd@NaGdF$_4$ (Gd-CS$_Y$SS in the absence of 20% Yb$^{3+}$). The lifetimes of Nd$^{3+}$ at 893 nm were measured under the 793 nm pulsed excitation, and the energy transfer efficiency of Nd$^{3+}$-to-Yb$^{3+}$ was calculated to be 79% (Fig. 4a). Similarly, to calculate the energy transfer efficiency of Yb$^{3+}$-to-Tm$^{3+}$, and to avoid the complex energy transfer pathways in the core-multishell structure, we produced a pair of simplified designs of NaGdF$_4$:20%Yb,1%Tm,29%Y@NaYF$_4$ (Gd-CS$_Y$ in the presence of 1% Tm$^{3+}$ energy acceptor) v.s. NaGdF$_4$: 20%Yb,30%Y@NaYF$_4$ (Gd-CS$_Y$ in the absence of Tm$^{3+}$). The 980 nm decay lifetimes of Yb$^{3+}$ were measured under the 920 nm pulsed excitation, and the energy transfer efficiency of Yb$^{3+}$-to-Tm$^{3+}$ was estimated to be 62% (Fig. 4b). To calculate the energy transfer efficieny of Tm$^{3+}$-to-Gd$^{3+}$, we produced a pair of samples of NaGdF$_4$:20%Yb,1%Tm,29%Y@NaYF$_4$ (Gd-CS$_Y$ in the presence of Gd$^{3+}$) v.s. NaYF$_4$: 20%Yb,1%Tm@NaYF$_4$ (Gd-CS$_Y$ in the absence of Gd$^{3+}$). By exciting the samples at 980 nm, the lifetimes of Tm$^{3+}$ at 290 nm were

measured and the energy transfer efficiency of Tm$^{3+}$-to-Gd$^{3+}$ was estimated to be 1% (Fig. 4c).

Furthermore, we conducted a quantitative study to compare the quantum yields of Gd-CS$_Y$SS and Gd-CS$_{Gd}$SS nanoparticles. The upconversion quantum yields from 240 to 750 nm of the as-prepared Gd-CS$_Y$S$_2$S$_3$ and Gd-CS$_{Gd}$S$_2$S$_3$ nanoparticles were estimated as 1.74 and 0.97%, respectively. To quantify the emission enhancement in the UV range from 240 to 325 nm, we also attempted to measure the upconversion quantum yields in the UV range, but without success due to the limited UVC emissions. Instead, we measured the quantum yields of upconversion emissions in the range from 240 to 400 nm, with the results being approximately 0.13 and 0.04%, respectively.

**The role of the first layer of NaYF$_4$ shell.** To probe the role of NaYF$_4$ layer in locking-in and recycling Gd$^{3+}$ excitation energy, we have compared the excited state lifetime of Gd$^{3+}$. As shown in Fig. 5 and Supplementary Fig. 15a significant prolonged (~4 times) lifetime of Gd$^{3+}$ emission from the $^6P_{7/2}$ level was achieved when the NaYF$_4$ first layer was applied. In contrast, there were negligible changes in the Gd$^{3+}$ lifetimes for emissions from $^6D_J$ and $^6I_J$ energy levels, indicating the energy loss from Gd$^{3+}$ to interior energy traps was mainly through $^6P_{7/2}$ energy level of Gd$^{3+}$ due to small energy gap between $^6D_J$, $^6I_J$, and $^6P_J$ (Supplementary Fig. 16). In addition, the emission intensities of Nd$^{3+}$ at 893 nm ($^4F_{3/2} \rightarrow {}^4I_{9/2}$), 1057 nm ($^4F_{3/2} \rightarrow {}^4I_{11/2}$), and 1330 nm ($^4F_{3/2} \rightarrow {}^4I_{13/2}$), and Tm$^{3+}$ at ~1460 nm ($^3H_4 \rightarrow {}^3F_4$) in the near-infrared range were essentially unaltered (Supplementary Fig. 17). These results indicate that the NaYF$_4$-assisted UCEL

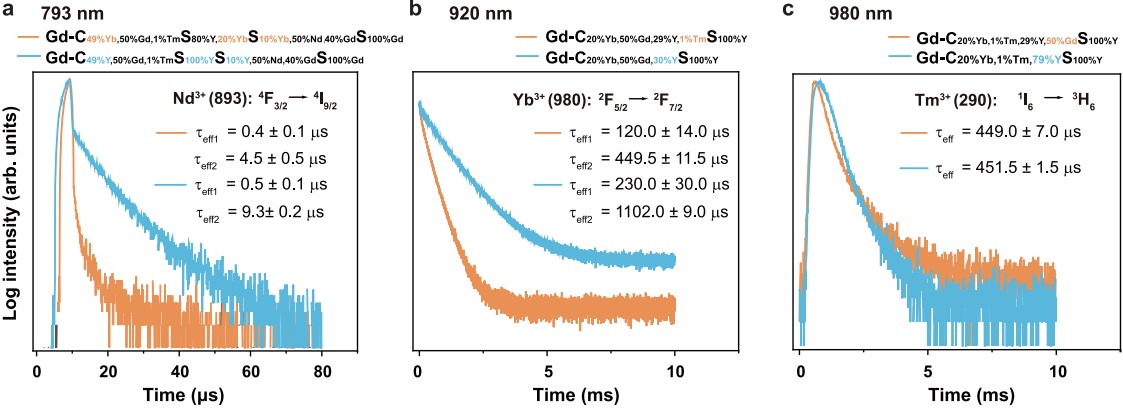

**Fig. 4 Lifetime measurements to quantify the step-wise energy transfer efficiencies of from Nd$^{3+}$ to Yb$^{3+}$, from Yb$^{3+}$ to Tm$^{3+}$ and from Tm$^{3+}$ to Gd$^{3+}$ ion. a** Luminescence decay curves of Nd$^{3+}$ emissions measured at 893 nm for NaGdF$_4$:49%Yb,1%Tm@NaYF$_4$:20%Yb@NaGdF$_4$:10%Yb,50%Nd@NaGdF$_4$ (with Yb$^{3+}$) and NaGdF$_4$:49%Y,1%Tm@NaYF$_4$@NaGdF$_4$:10%Y,50%Nd@NaGdF$_4$ (without Yb$^{3+}$) by pulsed 793 nm excitation. **b** Luminescence decay curves of Yb$^{3+}$ emissions measured at 980 nm for NaGdF$_4$:20%Yb,1%Tm,29%Y@NaYF$_4$ (with Tm$^{3+}$) and NaGdF$_4$: 20%Yb,30%Y@NaYF$_4$ (without Tm$^{3+}$) by pulsed 920 nm excitation. **c** Luminescence decay curves of Tm$^{3+}$ emissions measured at 290 nm for NaGdF$_4$:20%Yb,1%Tm,29%Y@NaYF$_4$ (with Gd$^{3+}$) and NaYF$_4$: 20%Yb,1%Tm@NaYF$_4$ (without Gd$^{3+}$) by pulsed 980 nm excitation.

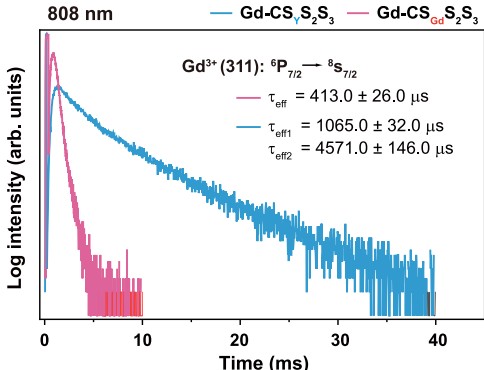

**Fig. 5 Lifetime decay analysis.** Upconversion luminescence decay curves of Gd$^{3+}$ emissions at 311 nm from Gd-CS$_Y$S$_2$S$_3$ v.s. Gd-CS$_{Gd}$S$_2$S$_3$ by pulsed 808 nm excitation.

mechanism favors the upconversion emissions from high-lying energy levels.

To further verify the role of NaYF$_4$ layer in enhancing the UVB and UVC emissions, we synthesized a group of Gd-CS$_Y$S$_2$S$_3$, NaGdF$_4$:49%Yb,1%Tm@NaGdF$_4$:20%Yb@NaYF$_4$:10% Yb,50%Nd@NaGdF$_4$ (Gd-CS$_1$S$_Y$S$_3$) and NaGdF$_4$:49%Yb,1% Tm@NaGdF$_4$:20%Yb@NaGdF$_4$:10%Yb,50%Nd@NaYF$_4$ (Gd-CS$_1$S$_2$S$_Y$) heterogeneous nanoparticles, in which NaGdF$_4$ was selectively replaced by NaYF$_4$ host lattice in the first, second and third layer, respectively (Fig. 6a). The intense UVB and UVC emission was only observed in Gd-CS$_Y$S$_2$S$_3$ nanoparticles. The emission profiles of Gd-CS$_1$S$_Y$S$_3$ and Gd-CS$_1$S$_2$S$_Y$ were quite similar to Gd-CS$_{Gd}$S$_2$S$_3$ nanoparticles. Moreover, when the half of optically inert Y$^{3+}$ ions in the first layer were replaced by the Gd$^{3+}$ ions, a drastic reduction of the Gd$^{3+}$ emission was observed, indicating that the NaYF$_4$ with Yb$^{3+}$ doping layer can effectively prevent the Gd$^{3+}$ energy leakage (Supplementary Fig. 18).

We further prepared a group of Gd-CS$_Y$S$_2$S$_3$ nanoparticles doped with Tb$^{3+}$ or Eu$^{3+}$ ions in the first layer NaGdF$_4$:49%Yb,1% Tm@NaYF$_4$:20%Yb,15%Tb@NaGdF$_4$:10%Yb,50%Nd@NaGdF$_4$ (Gd-CS$_{Y-15\%Tb}$S$_2$S$_3$) or NaGdF$_4$:49%Yb,1%Tm@NaYF$_4$:20%Yb,15% Eu@NaGdF$_4$:10%Yb,50%Nd@NaGdF$_4$ (Gd-CS$_{Y-15\%Eu}$S$_2$S$_3$), which can extract the excitation energy from Gd$^{3+}$ to emit green and red upconversion emissions through the scheme of energy migration

upconversion (EMU)[22]. Upon excitation at 808 nm, the characteristic emissions of Tb$^{3+}$ and Eu$^{3+}$ (highlighted in color) were observed (Fig. 6b, c and Supplementary Fig. 19), but no enhancement of UVB emissions observed. Doping with 15% Tb$^{3+}$ or Eu$^{3+}$ in the outmost layer only led to weak emission of Tb$^{3+}$ or Eu$^{3+}$ (Supplementary Fig. 20). The weak Tb$^{3+}$ and Eu$^{3+}$ emissions were attributed to the interior energy trapping of the excitation energy in the Gd$^{3+}$ sublattice. Together, these results indicate that an efficient energy transfer pathway (Nd$^{3+}$→ Yb$^{3+}$→Tm$^{3+}$→Gd$^{3+}$) occurs[50], and the excitation energy of Gd$^{3+}$ can be easily dissipated through the emission of Tb$^{3+}$, Eu$^{3+}$, or interior traps if without the first-shell layer of 20% Yb$^{3+}$ doped NaYF$_4$.

**Determination of the interior traps and Gd$^{3+}$ energy recycling above $^6P_J$.** The interior energy flux leakage pathway through lattice vibration and multiphonon transitions can be neglected because of the large energy gap of about 32 000 cm$^{-1}$ of Gd$^{3+}$ compared with the intrinsic low phonon energy of host materials (~350 cm$^{-1}$)[47]. Besides, it was reported that an efficient energy transfer can occur between Gd$^{3+}$ and Nd$^{3+}$ ions[51]. However, in our design, the energy transfer between these two ions did not happen. To preclude the possibility of the interior Nd$^{3+}$ energy trapping, we prepared a pair of Gd-CS$_{Gd}$S$_2$S$_3$ nanoparticles with and without Nd$^{3+}$ dopant NaGdF$_4$:49%Yb,1%Tm@NaGdF$_4$:20% Yb@NaGdF$_4$:10%Yb,50%Nd@NaGdF$_4$ and NaGdF$_4$:49%Yb,1% Tm @NaGdF$_4$:20%Yb@NaGdF$_4$:10%Yb,0%Nd@NaGdF$_4$ (Gd-CS$_{Gd}$S$_{50\%Nd}$S$_3$ and Gd-CS$_{Gd}$S$_{0\%Nd}$S$_3$). The lifetimes of Gd$^{3+}$ ($^6D_J$, $^6I_J$, $^6P_J$) and Tm$^{3+}$ ($^1I_6$, $^1D_2$) were virtually unchanged after removing Nd$^{3+}$ dopants in nanoparticles (Fig. 7a and Supplementary Fig. 21).

Since the Gd$^{3+}$–Gd$^{3+}$ energy migration is efficient and it can travel long distances[22], it is reasonable to assume that the excitation energy may be quenched by the interior lattice defects in the heterogenous structure with multi-layers of the shell. In our design, NaYF$_4$ in the first layer, effectively blocks the energy transfer from Gd$^{3+}$ to interior lattice defects in the outer shell layers, resulting in the migrating energy only recycling within the core domain of NaGdF$_4$:Yb,Tm (Fig. 7b). To validate our hypothesis, we investigated the lifetimes of Gd-CS$_Y$S$_2$S$_3$, Gd-CS$_1$S$_Y$S$_3$, and Gd-CS$_1$S$_2$S$_Y$ nanoparticles by changing the position of the NaYF$_4$ layer. As shown in Fig. 7c, the lifetime (Gd$^{3+}$: 311 nm) in Gd-CS$_Y$S$_2$S$_3$ is significantly longer than those in both

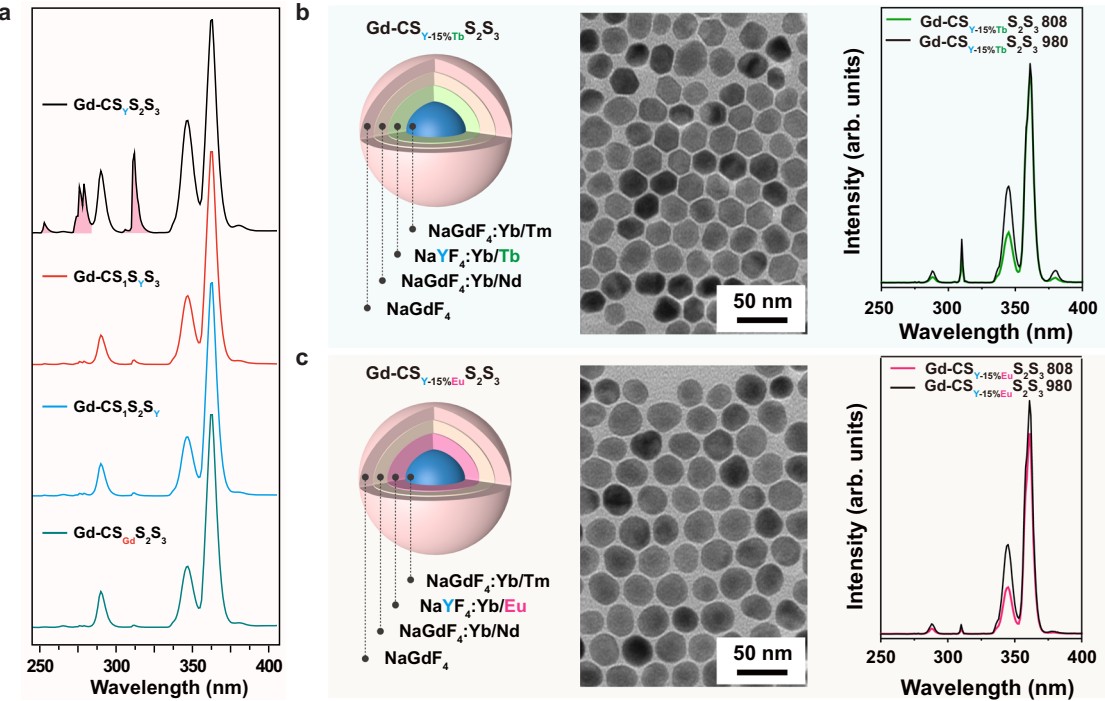

**Fig. 6 Characterization of core-multishell nanoparticles with alternative dopants and structural layouts. a** Room-temperature upconversion emission spectra of solutions containing Gd-CS$_Y$S$_2$S$_3$, Gd-CS$_1$S$_Y$S$_3$, Gd-CS$_1$S$_2$S$_Y$, and Gd-CS$_{Gd}$S$_2$S$_3$ nanoparticles under 808 nm excitation at a power density of 10 W cm$^{-2}$. **b, c** Schematic illustration, TEM images and photoluminescence spectra of the as-synthesized Gd-CS$_{Y-15\%Tb}$S$_2$S$_3$ and Gd-CS$_{Y-15\%Eu}$S$_2$S$_3$ nanoparticles.

Gd-CS$_1$S$_Y$S$_3$ and Gd-CS$_1$S$_2$S$_Y$ nanoparticles. The prolonged lifetime of Gd$^{3+}$ in Gd-CS$_Y$S$_2$S$_3$ nanoparticles is ascribed to the suppressed trapping of Gd$^{3+}$ energy by interior lattice defects in the multi-shell regions. By contrast, a similar level of short lifetimes of Gd$^{3+}$ in Gd-CS$_{Gd}$S$_2$S$_3$, Gd-CS$_1$S$_Y$S$_3$, and Gd-CS$_1$S$_2$S$_Y$ was observed, indicating that surface quenching was not responsible for the weak UV upconversion in conventional nanoparticles (Fig. 7d). This result is also consistent with our luminescence analysis in that a significantly stronger UV luminescence of Gd-CS$_Y$S$_2$S$_3$ nanoparticles compared with those of Gd-CS$_{Gd}$S$_2$S$_3$, Gd-CS$_1$S$_Y$S$_3$, and Gd-CS$_1$S$_2$S$_Y$ counterparts.

Furthermore, a Gd$^{3+}$ content of 50 mol% produced an optimum energy-migration property with a Gd$^{3+}$−Gd$^{3+}$ separation of 5.32 Å, which can be approximately calculated using the Eq. 3:[52]

$$d = \left(\frac{a^2 c\sqrt{3}/2}{1.5x}\right)^{1/3} \quad (3)$$

For the hexagonal NaGdF$_4$ unit cell, $a = 6.02$ Å, $c = 3.60$ Å. The short distance between Gd$^{3+}$ ions indicates that the Gd$^{3+}$−Gd$^{3+}$ energy migration is dominated by exchange interaction[46]. Moreover, it was reported by the Blasse group, $P_{em}$ for Gd$^{3+}$ is about $5 \times 10^2$ s$^{-1}$, $P_{(Gd^{3+} \to Gd^{3+})}$ is about $10^7$ s$^{-1}$. $P_{em}$ denotes the probability of emission, while $P_{(Gd^{3+} \to Gd^{3+})}$ denotes the probability for energy migration[46]. These results indicate that the excitation energy can be transferred more than $10^5$ times for the excited Gd$^{3+}$. After NaGdF$_4$ was replaced with NaYF$_4$ in the first layer, a significant increase in Gd$^{3+}$ lifetime was observed (Fig. 7c), suggesting the probability of Gd$^{3+}$-Gd$^{3+}$ energy migration within the core domain was significantly increased. Taken these together, the results conclusively suggested that the NaYF$_4$ shell can impede the fast migrating energy within the network of Gd$^{3+}$ ions from being trapped by the interior lattice defects in the outer multi-layer shell, which promotes the occurrence of energy hopping in Gd$^{3+}$ sublattice at the core domain of NaGdF$_4$:Yb,Tm, thereby realizing the intense UVB and UVC upconversion emissions.

**Enhancement in highly doped single nanoparticles**. To further evaluate UCEL mode in enhancing the high-order upconversion emissions in the heterogenous core-multishell structures, we implemented the similar design in the highly doped UCNP core, e.g., NaGdF$_4$:49%Yb,8%Tm@NaYF$_4$:20%Yb @NaGdF$_4$:10% Yb,50%Nd@NaYF$_4$ and NaGdF$_4$:49%Yb,8%Tm@NaGdF$_4$:20% Yb@NaGdF$_4$:10% Yb,50%Nd@NaGdF$_4$ (Gd-C$_{8\%Tm}$S$_Y$S$_2$S$_3$ and Gd-C$_{8\%Tm}$S$_{Gd}$S$_2$S$_3$), and quantify the brightness of single UCNPs using a purpose-built confocal microscopy system (Supplementary Fig. 22). Due to the significant UV absorption by the optical components, including the objective lens and mirrors, instead of a direct quantification of the UVC emissions at a single nanoparticle level, we monitored the amount of the blue band emissions from a single nanoparticle. Under the same excitation power from both 808 nm and ~980 nm lasers, the emission intensities of Gd-C$_{1\%Tm}$S$_Y$S$_2$S$_3$ and Gd-C$_{1\%Tm}$S$_{Gd}$S$_2$S$_3$ nanoparticles under the 808 nm excitation were ~4 times and ~5 times higher than those under the ~980 nm excitation, respectively (Supplementary Fig. 23). In contrast, much higher enhancement factors of the highly doped Gd-C$_{8\%Tm}$S$_Y$S$_2$S$_3$ (~25 times) and Gd-C$_{8\%Tm}$S$_{Gd}$S$_2$S$_3$ (~15 times) nanoparticles were achieved under the 808 nm v.s. ~980 nm excitations. These results suggest UCEL mode could be broadly applied to a variety of UCNP core concentrations[38] and under a large dynamic range of excitation power densities[53], suitable for both ensemble and single nanoparticle applications[54].

**Potential in enhancing Reactive Oxygen Species (ROS) generation**. Moreover, we prepared the titanium dioxide (TiO$_2$)-coated UCNPs in which TiO$_2$ serves as the photosensitizer[55].

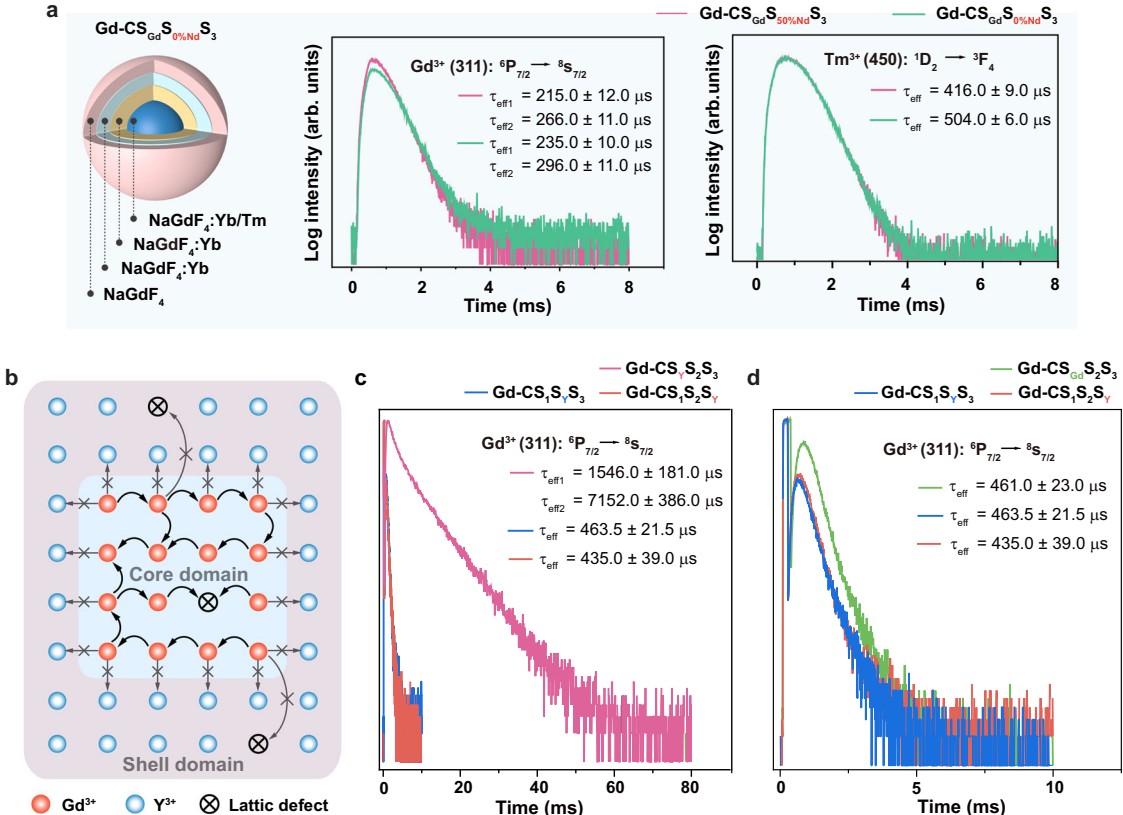

**Fig. 7 Mechanistic investigation of $Gd^{3+}$ energy recycling to avoid the interior traps in the heterogeneous core-shell nanoparticle. a** Schematic illustration of the as-synthesized $Gd\text{-}CS_{Gd}S_{0\%Nd}S_3$ nanoparticles, and the upconversion luminescence decay curves of $Gd^{3+}$ emission at 311 nm and $Tm^{3+}$ emission at 450 nm of $Gd\text{-}CS_{Gd}S_{50\%Nd}S_3$ and $Gd\text{-}CS_{Gd}S_{0\%Nd}S_3$ nanoparticles under 980 nm excitation, respectively. **b** Schematic illustration of energy recycling in $Gd^{3+}$ sublattice at the core domain of $NaGdF_4$:Yb,Tm. **c** The upconversion luminescence decay curves of $Gd^{3+}$ emission at 311 nm in $Gd\text{-}CS_YS_2S_3$, $Gd\text{-}CS_1S_YS_3$, and $Gd\text{-}CS_1S_2S_Y$ nanoparticles by the pulsed 808 nm excitation. **d** The upconversion luminescence decay curves of $Gd^{3+}$ emission at 311 nm in $Gd\text{-}CS_{Gd}S_2S_3$, $Gd\text{-}CS_1S_YS_3$, and $Gd\text{-}CS_1S_2S_Y$ nanoparticles by the pulsed 808 nm excitation.

TEM and X-ray powder diffraction (XRD) analysis confirmed the successful synthesis (Supplementary Fig. 24), and compositional analysis of these nanocomposites by energy-dispersive X-ray spectroscopy (EDX) confirms the presence of $Ti^{4+}$, $Nd^{3+}$, $Gd^{3+}$, $Yb^{3+}$, and $Tm^{3+}$ (Supplementary Fig. 25). As shown in Supplementary Fig. 26, the emission $Gd\text{-}CS_YS_2S_3@TiO_2$ became weaker compared with $Gd\text{-}CS_YS_2S_3$ nanoparticles due to the absorbance of UV emission by the $TiO_2$ shell. The ability to generate singlet oxygen ($^1O_2$) of the as-synthesized nanocomposites was evaluated by the 1,3-diphenylisobenzofuran (DPBF) chemical probe under 808 nm laser irradiation. The characteristic absorbance of DPBF gradually decreased with the increase in irradiation time, the characteristic absorption decreased with time, indicating the successful generation of $^1O_2$ (Supplementary Fig. 26b). These results indicate the enticing prospects of NIR light-mediated photosensitizing nanocomposites for ROS generation and their potential applications in photocatalysis and biomedical fields.

## Discussion

In this study, we demonstrated a UCEL approach through the core-multishell heterogeneous structure design to regulate the energy transfer pathway in lanthanide-doped UCNPs for UVC and UVB generation by 808 nm excitation. The key to this design is the utilization of an optical inert $NaYF_4$ host lattice with $Yb^{3+}$ doping as an interlayer between the multiple cascade NIR photon sensitization shells and upconversion emitting core. Therefore, the sensitized NIR excitation energies can be transferred inbound and upconverted at the core domain of $NaGdF_4$:Yb,Tm, where

high-concentration $Gd^{3+}$ ions can recycle among the higher-lying excited energy states above $^6P_J$ to realize intense UVB and UVC upconversion emissions. We believe this approach will advance the design rationale for enhancing the NIR sensitized UV upconversion emissions towards the potential areas of biomedicine, information technology, photocatalysis, environmental science, and many other emerging fields.

## Methods

**Nanoparticles synthesis.** We synthesized the core–multishell nanoparticles using the method described in ref. [24]. and ref. [40]. Additional experimental details are provided in the Supplementary Note.

**Synthesis of UCNPs@TiO$_2$ nanocomposites.** $Gd\text{-}CS_YS_2S_3@TiO_2$ nanocomposites were synthesized according to a modified literature procedure[55,56]. Typically, 66 mg/mL (0.3 mL) as-prepared oleic acid nanoparticles $Gd\text{-}CS_YS_2S_3$ were dispersed in a 0.2 M HCl solution followed by ultrasonication to remove the surface ligands. After that, ligand-free UCNPs were collected via centrifugation. The ligand-free nanoparticles were washed with deionized water and ethanol several times, and then dispersed in 4 mL of deionized water containing 0.8 g poly-vinylpyrolidone (average Mw 40 000) with ultrasonication and stirring for 1 h. Then, 20 mL ethanol was added under magnetic stirring for 30 min. $TiF_4$ aqueous solution (2.4 mL 0.025 M) was dropwise added into the solution under stirring. Then the whole solution was transferred into a 50 mL Teflon-lines autoclave and heated at 180 °C for 4 h. After cooling to the room temperature, the as-prepared products were collected by centrifugation, washed with deionized water and ethanol several times, and dried at 65 °C.

**Single particle imaging.** The emission intensities of single nanoparticles were recorded using a laboratory-built confocal microscopy system. Supplementary Fig. 25 shows the schematic drawing of the experimental setup, where UCNPs are excited by a polarization-maintaining single-mode fiber-coupled ~980 nm

(BL976-PAG900, controller CLD1015, Thorlabs) or 808 nm (F280APC-808, Leoptics) diode lasers. The first half-wave plate (HWP, WPH05M-980, Thorlabs) and a polarized beam splitter (PBS, CCM1-PBS252/M, Thorlabs) are employed to control the excitation power by rotating HWP electronically. The purpose of the second HWP is to turn the polarization from horizontal to vertical. The same setup is used for the 808 nm laser excitation, combined to the ~980 nm excitation path by the first dichroic mirror (DM, T842lp, Chroma). After collimation, the excitation beam is reflected by the short-pass dichroic mirror (DM, T785spxrxt-UF1, Chroma), and focused through a high numerical aperture objective (UPlanSApo, 100×/1.40 oil, Olympus) to the sample slide. Photoluminescence is collected by the same objective and split from the excitation beams by a dichroic mirror DM. The emission signals are filtered by a short pass filter (SPF, FF01-750SP, Semrock), coupled into a multimode fiber (MMF, M42L02, Thorlabs), and detected by a single-photon counting avalanche photodiode (SPAD, SPCM-AQRH-14-FC, Excelitas). The MMF can also be switched to a monochromator (iHR550, Horiba) for upconversion emission spectrum measurement.

**Evaluation of singlet oxygen generation**. The chemical probe 1,3-diphenyliso-benzofuran (DPBF) can be used to evaluate the amount of produced singlet oxygen ($^1O_2$) from the as-prepared Gd-CS$_Y$S$_2$S$_3$@TiO$_2$ under 808 nm laser irradiation. DPBF can react with singlet oxygen ($^1O_2$) irreversibly and then cause the intensity decrease of its characteristic absorption at 417 nm[55]. Therefore, the amount of $^1O_2$ produced under laser irradiation can be evaluated by the absorption signal of DPBF with a UV−Vis absorption spectrum.

**Caculation of absorption cross-section σ of Nd$^{3+}$**. The approximate absorption cross-section σ of Nd$^{3+}$ at 808 nm was calculated from the UV−Vis absorption spectra of the nanoparticles using the following equations[44]:

$$A = \varepsilon \cdot M \cdot l \tag{4}$$

$$\sigma = \frac{\varepsilon}{n} \tag{5}$$

where $A$ is the absorbance, $\varepsilon$ is the absorption coefficient, $M$ is the molar concentration of Nd$^{3+}$, $l$ is the path length, $n$ is the atomic number density of Nd$^{3+}$ ions. $\sigma = 1.5 \times 10^{-19}$ cm$^2$ (808 nm for Gd-CS$_Y$S$_2$S$_3$), $\sigma = 1.3 \times 10^{-19}$ cm$^2$ (808 nm for Gd-CS$_{Gd}$S$_2$S$_3$).

## Data availability

All the relevant data are available from the correspondence authors upon reasonable request. Source data are provided with this paper.

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

## Acknowledgements

The authors thank the National Basic Research Program of China (No. 2016YFA0201600), the National Natural Science Foundation of China (Nos. 21701109 and 31771105), Shanghai Shuguang Program (18SG29) and Natural Science Foundation of Shanghai (18ZR1401700) for financial supports. The authors acknowledge the assistance of SUSTech Core Research Facilities. The authors thank Prof. X. Liu, Prof. F. Li, Prof. F. Wang, Prof. K. Zheng, Prof. X. Zhu, Prof. A. Cao, Dr. S. Han and Dr. J. Zhou for helpful discussions. The authors thank Prof. W. Feng and Y. Cai for their help with lifetime measurements. The authors thank R. Liu for her help with quantum field measurements.

## Author contributions

Q. S. and D. J. conceived the project. Q. S., H. W., and D. J. designed the experiments and supervised the research. Q. S., H. L. W., Y. L., C. C., M. G., S. W., Y. S., and Z. C. were primarily responsible for the experiments of nanoparticles synthesis and characterization. Q. S., D. J., H. L. W., Y. L., C. C., Y. S., and S. W. contributed to the data analyses and discussion. Q. S. and H. L. W. prepared the figures. Q. S. and D. J. wrote the paper with input from other authors.

## Competing interests

The authors declare no competing interests.
