## [Peer Review File · Nature Communications]

REVIEWER COMMENTS

Reviewer #1 (Remarks to the Author):

The work of Su et al. reports a six-photon upconversion process in Gd³⁺/Tm³⁺-codoped nanoparticles comprising a heterogeneous core-multishell nanostructure. Near infrared-to-deep ultraviolet upconversion is an interesting topic in upconverting nanoparticles with potential applications in drug release. Although this interest and the challenges to be overcome to develop 800 nm-sensitized UCNPs are relatively well discussed in the introduction, the novelty of this paper is not sufficiently highlighted.

Furthermore, the conclusions of the paper supporting the existence of an upconverted excitation lock-in mode that effectively suppresses energy dissipation by interior traps are essentially based on qualitative arguments. The authors argue that the key point of the manuscript is the design of the nanostructure where "the NaYF₄-based first shell layer selectively blocks the energy transfer from Gd³⁺ to interior energy traps (e.g., lattice defects and impurities)." It is assumed that this layer "preserves and recycles the excitation energy within the core region, leading to increased populations in the ⁶D_J, ⁶I_J, and ⁶P_J states of Gd³⁺ and intense UV emission of Gd³⁺." This assumption is based only on empirical evidence and to justify publication in Nature Communications a quantitative proof involving ion-ion energy transfer calculations is required (for instance to support quantitatively that "the sensitized NIR excitation energies can be transferred inbound and upconverted at the core domain of NaGdF₄:Yb,Tm, where high-concentration Gd³⁺ ions can recycle among the higher-lying excited energy states above ⁶P_J to realize intense deep UV upconversion emissions").

In summary, although the paper has the potential to attract the attention of the journal readers, it requires a quantitative and deep description of all the energy transfer processes to validate the empirical assumptions made justifying, thus, publications in Nature Communications.

Other aspects requiring an improvement:

- 1) What are the solvent and the volume fraction in the solution of the NPs shown in Fig. 5? Is the efficiency of radiation-to-heat conversion similar to the two kinds of particles? (note that light is strictly used for radiation emitted in the visible part of the electromagnetic spectrum).
- 2) What is the meaning of the full lines in Fig. S1? What function is used to fit the size distributions?
- 3) It is difficult to grasp the message from Figs. S9 and S10b (all the bars have the same height...)
- 4) What are the experimental conditions of the absorption spectra shown in Fig. S10 (solvent, NPs volume fraction, reference, etc)? To support the sentence "(...) although the absorption profile of Gd-CSYS₂S₃ is not changed compared with that of Gd-CSGdS₂S₃ nanoparticles (Supplementary Fig. S11)" and, thus, validate the qualitative comparison of intensities between different samples authors must record their absorbance in absolute units (not arb. units). Otherwise, the measurement of emission absolute quantum yields is mandatory.
- 5) A short paragraph reviewing the work performed in Nd³⁺-sensitized UCNPs must be added.

Reviewer #2 (Remarks to the Author):

This is an interesting paper in which authors have designed and synthesized a new type of core-shell structure capable of intense UV emission (below 300 nm). For that they have used a complex structure in which the presence of Gd ions serve as energy reservoir. The idea itself is interesting. The materials prepared are of outstanding quality. The text is well-written and the figures have been nicely prepared and designed.

The final result is a nanoparticle capable of UV emission under 808 nm radiation. At the beginning of their manuscript authors stated "Deep-UV upconversion emission is particularly useful for drug release in deep tissues...." I think that the paper needs such demonstration. I recognise the impact of the paper and its relevance. But it is not demonstrated that the resulting nanostructures can have a practical application. In particular, it would be nice to demonstrate the application of these structures where other have failed. I think the impact and quality of the paper could be improved by orders of magnitude by including some evidence of the advances that can be achieved by such structures.

In connection with the previous comment, authors are claiming that their design improves the final UV emission intensity. They compare in the main text the emission spectra in presence and absence of Gd showing the UV emission improvement. But from that figure it is difficult to have a quantitative idea of how big is the improvement. I think the paper will benefit from a precise qualification of the improvement achieved. Also if a number is given in main text about QY, some comparison with previous results are required.

Finally, the discussion about thermal loading effects is confusing. All along the work authors state that they use 808 nm (Nd) absorption to avoid the heating of 980 nm radiation (Yb) but then data included in the figure 5 seems to indicate the opposite. In general this figure and related discussion should be re-designed to make it clear. Also, the analysis of the heating as a function of pump power should be included to evaluate the different heat pathways that are different from the one include in Figure 5.

In summary, it is a good piece of work and I think the authors can make it even better and suitable for Nature Comm if they include some new experiments to show the practical potential of the structure as well as to quantify the improvement they have achieved.

Reviewer #3 (Remarks to the Author):

Most upconversion systems developed so far concentrate in converting near-infrared light into visible and near UV spectral ranges. However, light with shorter wavelength, in the UV-C range has many industrial and medical applications. Here the authors propose a sophisticated type of core-multishell upconverting nanoparticles which could potentially be provided for in situ production of singlet oxygen and/or drug release in photodynamic treatment of cancer.

This described work is an expansion of a study published last year in *Nanoscale* (ref. 30) in which Gd-C(YbTm)S(Yb)S(YbNd)S(Gd) nanoparticles, based on the NaGdF₄ host, were described and showed to have more intense Tm emission under 808-nm (Nd) excitation compared with 980-nm (Yb) excitation; additionally transformation in light onto heat was also demonstrated. In the present work, the authors replace the host material of the first shell, NaGdF₄ with optically inactive NaYF₄, while keeping all dopant concentrations the same. This simple modification has a dramatic effect on the photophysical properties in that now Gd ions can be excited by a 6-photon process.

The experimental section is convincing and all necessary control experiments have been performed, so that the result is well proved. The paper demonstrates that seemingly minor, but well thought, modifications of hetero nanostructures can lead to substantial beneficial

modifications of their photophysical properties and opens large perspectives in the field.

I am missing some more quantitative aspects, regarding efficiencies of energy transfer and overall quantum yield for instance; uncertainties on reported data are also missing as well as the number of repeat experiments made.

Remarks

a. I question the use of "deep UV" for the spectral range of the upconverted wavelength produced by the described core-multi-shell nanoparticles. This denomination is too vague. In classical nomenclature, the UV range (400-10 nm) is subdivided between near UV (400-300 nm), middle UV (300-200 nm), and far UV (200-10 nm; sometimes subdivided in far UV – 200-100 nm and extreme UV – 100-10 nm). Therefore since the upconverted light described in this paper has wavelength around 250 nm, "deep UV" should be replaced with "middle UV". Alternatively, the authors could use the other classification, namely UV-C (100-290 nm).

b. I am puzzled by the use of "optically inactive shell" or "optical inert interlayer" for the first NaYF₄ shell because its composition is NaYF₄:Yb(20%) and it acts as an energy migrator relay, so the shell is not optically inactive, but the host matrix is. I do not quite understand. Is the first shell homogeneous, i.e. entirely comprised of NaYF₄:Yb(20%) or is it heterogeneous with a first undoped layer? Please clarify.

c. Figure 1, caption. Please replace "6-photon" with "6th photon" since the transfer from Tm to Gd is a one-photon excitation.

d. Results. First section. The description of the phenomena occurring in the nanoparticles lacks precision. For instance in the third sentence why describing energy transfer to Tm as being a back energy transfer? Back transfer from Yb would be transfer to Nd. Fourth sentence: Why write "By doping of Gd" since Gd is in the host material?

e. Results, second section. It would be clearer for the reader to give the composition of the nanoparticles used in this work rather than the one of those described in ref. 30, as is done in the "controlled synthesis" section.

f. Figure 2d. I am surprised by the semi-log plot. Usually a log-log plot should be used to determine the number of implied photons, as one can easily imply from the formula given on page 6 (2nd line).

Responses to the reviewers' comments (Manuscript number: NCOMMS-20-39080 for Nature Communications)

Title: Upconverted excitation energy lock-in for deep-ultraviolet enhancement

For editor and reviewers' convenience, we have listed below the new experiments and major advances made in the revision:

As suggested by reviewer #1, we have carried out additional experiments to evaluate the energy transfer efficiency and quantum yields of as-prepared nanoparticles. Experimental results indicate that a significantly enhanced emission intensity of our heterogenous core-multishell nanoparticles was obtained relative to that of conventional Gd-based counterparts (Figs. 3, S14, R1-3). We re-designed the experiments to determine that the traps are interior lattice defects (Fig. 6b-d). The radiation-to-heat conversion (Fig. R4) and the qualitative comparison of their absorbance (Fig. S11) were also conducted. Moreover, as suggested by the reviewer, we further discussed the energy transfer processes, energy recycling in high concentration Gd³⁺ core domain to validate our empirical assumptions.

As suggested by reviewer #2, we have performed the comparative quantum yield studies for our heterogenous core-multishell nanoparticles and conventional Gd-based counterparts (Figs. 3, S14, R1-3). Thermal loading effects experiments were redesigned and shown in Fig. R6. To demonstrate our nanoparticles' potential in ROS initiated applications, we prepared titanium dioxide (TiO₂)-coated upconversion nanoparticles and evaluated the generation of ROS induced by our upconversion composites (Figs. S24-26).

To address the issues raised by reviewer #3, we investigated the efficiency of energy transfer process and overall quantum yield of the two kinds of nanoparticles (Figs. 3, S14, R1-3). The log-log plots have been provided to determine the 6 and 5 photon upconversion processes (Figs. 2g and S13). We have also added uncertainties on reported data and the number of repeat experiments made.

Reviewer #1 (Remarks to the Author):

Comment #1: The work of Su et al. reports a six-photon upconversion process in Gd³⁺/Tm³⁺-codoped nanoparticles comprising a heterogeneous core-multishell nanostructure. Near infrared-to-deep ultraviolet upconversion is an interesting topic in upconverting nanoparticles with potential applications in drug release. Although this interest and the challenges to be overcome to develop 800 nm-sensitized UCNPs are relatively well discussed in the introduction, the novelty of this paper is not sufficiently highlighted.

Response: We thank the reviewer for the positive evaluation and suggestion on our work. As suggested, we have highlighted the novelty of this paper in the revised manuscript. Apart from the intrinsic parity-forbidden nature of 4f–4f optical transitions in lanthanide systems, the typically low conversion efficiency arises mainly from the presences of lattice defects (surface defects) and surface molecules, which leads to unwanted nonradiative energy transfer between dopants and quenchers [ACS Cent. Sci. **2019**, 5, 29; J. Am. Chem. Soc. **2017**, 139, 3275; Advanced Science **2021**, 8 (6), 2003325]. Therefore, present studies on six-photon upconversion are mainly focused on core–multishell nanoparticles comprising spatially confined dopant ions. Although lattice defects include both surface defects and interior defects, to our best knowledge, little attention has been paid to the effect of interior defects on upconversion luminescence [Angew. Chem. Int. Ed. **2017**, 56, 7605, Advanced Science **2021**, 8 (6), 2003325]. In this study, we utilized a heterogeneous core-multishell nanostructure to suppress the energy transfer from lanthanide to interior defects, which resulted in an unusual six-photon upconverted UV emission at 253 nm under 808 nm excitation. Our mechanistic investigation reveals upconverted excitation lock-in (UCEL) mode in which Gd³⁺-sensitized excitation energy can be efficiently retained and be released in the form of emission from ⁶D_J, ⁶I_J, and ⁶P_J states of Gd³⁺.

Comment #2: Furthermore, the conclusions of the paper supporting the existence of an upconverted excitation lock-in mode that effectively suppresses energy dissipation by interior traps are essentially based on qualitative arguments. The authors argue that the key point of the manuscript is the design of the nanostructure where “the NaYF₄-based first shell layer selectively blocks the energy transfer from Gd³⁺ to interior energy traps (e.g., lattice defects and impurities).” It is assumed that this layer “preserves and recycles the excitation energy within the core region, leading to increased populations in the ⁶D_J, ⁶I_J, and ⁶P_J states of Gd³⁺ and intense UV emission of Gd³⁺.” This assumption is based only on empirical evidence and to justify publication in Nature Communications a quantitative proof involving ion-ion energy transfer calculations is required (for instance to support quantitatively that “the sensitized NIR excitation energies can be transferred inbound and upconverted at the core domain of NaGdF₄:Yb,Tm, where high-concentration Gd³⁺ ions can recycle among the higher-lying excited energy states above ⁶P_J to realize intense deep UV upconversion emissions”).

In summary, although the paper has the potential to attract the attention of the journal readers, it requires a quantitative and deep description of all the energy transfer processes to validate the empirical assumptions made justifying, thus, publications in Nature Communications.

Response: We thank the reviewer’s insightful and valuable comments on our manuscript. We have seriously taken this reviewer’s comments and conducted the quantitative studies on the efficiencies of all the energy transfer processes. The large energy gap of about 32000 cm⁻¹ of Gd³⁺ and intrinsic low phonon energy of NaGdF₄ offer good possibilities to obtain 100% energy transfer efficiency from Gd³⁺-to-Gd³⁺ [Mater. Chem. Phys. **1987**, 16, 201; Blasse, G.; Grabmaier, B. C. Luminescent Materials;

Springer: Berlin, 1994. P100]. The energy transfer efficiencies η of Nd³⁺-to-Yb³⁺, Yb³⁺-to-Tm³⁺, and Tm³⁺-to-Gd³⁺ have been primarily estimated from eq 1 and eq 2 [*Phys. Rev. B* **1971**, *4*, 3153; *J. Lumin.* **2015**, *166*, 177]:

$$\eta = 1 - \frac{\tau_m}{\tau_{Ln}} \quad (1)$$

$$\tau_m = \frac{\sum \alpha_i \tau_i^2}{\sum \alpha_i \tau_i} \quad (2)$$

Where τ_m is the mean lifetime of energy donor lanthanides (Ln) in the presence of energy acceptor, τ_{Ln} is the intrinsic lifetime of energy donor, and α is the amplitude. To calculate the energy transfer efficiencies of Nd³⁺-to-Yb³⁺, Yb³⁺-to-Tm³⁺, and Tm³⁺-to-Gd³⁺, we designed and synthesized three pairs of heterogeneous nanoparticles (TEM results shown in Supplementary Fig. 14). In our experiment, to first determine the intrinsic lifetime of the corresponding energy donors, the energy acceptors of Yb³⁺, Tm³⁺, and Gd³⁺ were replaced by optically inert Y³⁺ ions.

In details, to calculate the energy transfer efficiency of Nd³⁺-to-Yb³⁺, we produced a pair of samples of 49%Yb,50%Gd,1%Tm@80%Y,20%Yb@10%Yb,50%Nd,40%Gd@100%Gd (Gd-CS_YSS in the presence of 20% Yb³⁺ energy acceptor) v.s. 49%Y,50%Gd,1%Tm@100%Y@10%Y,50%Nd,40%Gd@100%Gd (Gd-CS_YSS in the absence of 20% Yb³⁺). The lifetimes of Nd³⁺ at 893 nm were measured under the 793 nm pulsed excitation, and the energy transfer efficiency of Nd³⁺-to-Yb³⁺ was calculated to be 79% (Figure 3a). Similarly, to calculate the energy transfer efficiency of Yb³⁺-to-Tm³⁺, and to avoid the complex energy transfer pathways in the core-multishell structure (Figure R2), we produced a pair of simplified designs of 50%Gd,20%Yb,1%Tm,29%Y@100%Y (Gd-CS_Y in the presence of 1% Tm³⁺ energy acceptor) v.s. 50%Gd,20%Yb,30%Y@100%Y (Gd-CS_Y in the absence of Tm³⁺). The 980 nm decay lifetimes of Yb³⁺ were measured under the 920 nm pulsed excitation, and the energy transfer efficiency of Yb³⁺-to-Tm³⁺ was estimated to be 62% (Figure 3b). To calculate the energy transfer efficiency of Tm³⁺-to-Gd³⁺, we produced a pair of samples of 50%Gd,20%Yb,1%Tm,29%Y@100%Y (Gd-CS_Y in the presence of Gd³⁺) v.s. 79%Y,20%Yb,1%Tm@100%Y (Gd-CS_Y in the absence of Gd³⁺). By exciting the samples at 980 nm, the lifetimes of Tm³⁺ at 290 nm were and the energy transfer efficiency of Tm³⁺-to-Gd³⁺ was estimated to be 1% (Figure 3c).

Figure 3 | Energy transfer efficiency of Nd³⁺-to-Yb³⁺, Yb³⁺-to-Tm³⁺ and Tm³⁺-to-Gd³⁺ ion. (a) Luminescence decay curves of Nd³⁺ emissions measured at 893 nm for 49%Yb,50%Gd,1%Tm@80%Y,20%Yb@10%Yb,50%Nd,40%Gd@100%Gd (with Yb³⁺) and 49%Y,50%Gd,1%Tm@100%Y@10%Y,50%Nd,40%Gd@100%Gd (without Yb³⁺) by pulsed 793 nm

excitation. (b) Luminescence decay curves of Yb^{3+} emissions measured at 980 nm for 50%Gd,20%Yb,1%Tm,29%Y@100%Y (with Tm^{3+}) and 50%Gd,20%Yb,30%Y@100%Y (without Tm^{3+}) by pulsed 920 nm excitation. (c) Luminescence decay curves of Tm^{3+} emissions measured at 290 nm for 50%Gd,20%Yb,1%Tm,29%Y@100%Y (with Gd^{3+}) and 79%Y,20%Yb,1%Tm@100%Y (without Gd^{3+}) by pulsed 980 nm excitation.

It should be noted that when calculating the energy transfer efficiency of Yb^{3+} -to- Tm^{3+} and Tm^{3+} -to- Gd^{3+} , we began with the synthesis of 49%Yb,50%Gd,1%Tm@80%Y,20%Yb@10%Yb,50%Nd,40%Gd@100%Gd (Gd- CS_YSS with Tm^{3+}) v.s. 49%Yb,50%Gd,1%Y@80%Y,20%Yb@10%Yb,50%Nd,40%Gd@100%Gd (Gd- CS_YSS without Tm^{3+}), and 49%Yb,50%Gd,1%Tm@80%Y,20%Yb@10%Yb,50%Nd,40%Gd@100%Gd (Gd- CS_YSS with Gd^{3+}) v.s. 49%Yb,50%Y,1%Tm@80%Y,20%Yb@10%Yb,50%Nd,40%Gd@100%Gd nanoparticles (Gd- CS_YSS without Gd^{3+}) which have the same structure as our research. However, the lifetime of Yb^{3+} for Gd- CS_YSS with Tm^{3+} is longer than that of Gd- CS_YSS without Tm^{3+} , indicating the inaccurate results caused by the complex energy transfer pathways in the core-multishell structure (Figure R1a). Similarly, with Gd^{3+} in the core nanoparticles, we observed longer lifetime of Tm^{3+} at 290 nm compared with that of their counterparts without Gd^{3+} (Figure R1b). Then, we prepared a simplified 49%Yb,50%Gd,1%Y@100%Y (Gd- CS_Y with 49% Yb^{3+}) v.s. 49%Yb,50%Gd,1%Tm@100%Y (Gd- CS_Y with 49% Yb^{3+} and 1% Tm^{3+}) and 49%Yb,50%Gd,1%Tm@100%Y (Gd- CS_Y with 49% Yb^{3+} and 50% Gd^{3+}) v.s. 49%Yb,50%Y,1%Tm@100%Y (Gd- CS_Y with 49% Yb^{3+}) core-shell nanoparticles. These two pairs of nanoparticles could not be utilized to calculate the energy transfer efficiency of Yb^{3+} -to- Tm^{3+} as well (Figure R2), which can be attributed to the excitation energy dissipation through nonradiative process in concentrated Yb^{3+} sublattice [Angew. Chem. Int. Ed. 2019, 58, 17255; Adv. Sci. 2021, 2003325]. Besides, we attempted to directly excite Tm^{3+} ion to measure the energy transfer efficiency of Tm^{3+} -to- Gd^{3+} by using pulsed 778 nm laser. The upconversion emission of Tm^{3+} at 290 nm and their lifetime could hardly be spectroscopically detected due to their small absorption cross-section (Figure R3).

Figure R1. Lifetime decay analysis. (a) Lifetime decay analysis of Yb^{3+} emission at 980 nm for 49%Yb,50%Gd,1%Tm@80%Y,20%Yb@10%Yb,50%Nd,40%Gd@100%Gd (with Tm^{3+}) v.s. 49%Yb,50%Gd,1%Y@80%Y,20%Yb@10%Yb,50%Nd,40%Gd@100%Gd (without Tm^{3+}) by pulsed 920 nm excitation. (b) Lifetime decay analysis of Tm^{3+} emissions at 290 nm for 49%Yb,50%Gd,1%Tm@80%Y,20%Yb@10%Yb,50%Nd,40%Gd@100%Gd (with Gd^{3+}) v.s. 49%Yb,50%Y,1%Tm@80%Y,20%Yb@10%Yb,50%Nd,40%Gd@100%Gd (without Gd^{3+}) by pulsed 980 nm excitation.

49%Yb,50%Y,1%Tm@80%Y,20%Yb@10%Yb,50%Nd,40%Gd@100%Gd (without Gd³⁺) by pulsed 808 nm excitation.

Figure R2. Lifetime decay analysis. (a) Lifetime decay analysis of Yb³⁺ emissions at 980 nm for 49%Yb,50%Gd,1%Tm@100%Y (with Tm³⁺) v.s. 49%Yb,50%Gd,1%Y@100%Y (without Tm³⁺) by pulsed 920 nm excitation. (b) Lifetime decay analysis of Tm³⁺ emissions at 290 nm from 49%Yb,50%Gd,1%Tm@100%Y (with Gd³⁺) v.s. 49%Yb,50%Y,1%Tm@100%Y (without Gd³⁺) by pulsed 980 nm excitation.

Figure R3. Emission profile of as-prepared nanoparticles with different nanostructures under excitation of 778 nm.

We further thoroughly discussed all the energy transfer processes to validate the empirical assumptions. As shown in Figure 1 in revised manuscript, the 808 nm photons are first sensitized by

Nd³⁺ sensitizer ions, being populated at the ⁴F_{5/2} energy state and quickly relaxed to the ⁴F_{3/2} energy state of Nd³⁺. The excited Yb³⁺ ions serve as an energy migrator to sensitize and pass on the energy from Nd³⁺ and to populate the ³P₂ state of Tm³⁺ through a five-photon upconversion process. Subsequently, the energy at the ³P₂ state relax non-radiatively to populate ¹I₆ and give rise to UVB emissions at 290 nm. Besides, Gd³⁺ ions extract the excitation energy through the energy transfer process of ¹I₆→³H₆ (Tm³⁺): ⁸S_{7/2}→⁶P_J (Gd³⁺). The excitation energy of Gd³⁺ at ⁶P_J can resist nonradiative quenching due to its large energy gap (~32 200 cm⁻¹ from ⁶P_J to ⁸S_{7/2}). Thus, the lifetime of Gd³⁺ at ⁶P_J energy state is long enough for the sixth photon to be absorbed from the excited Yb³⁺. Therefore, the ⁶D_J state of Gd³⁺ is further populated by the appropriate energy matching of the following transitions of ²F_{5/2}→²F_{7/2} (9750 cm⁻¹, Yb³⁺): ⁶P_J→⁶D_J (~8750 cm⁻¹, Gd³⁺) [*Opt. Lett.* **2008**, 33, 19; *J. Phys. Chem. Lett.* **2015**, 6, 556; *Light Sci. Appl.* **2014**, 3, e193]. Thus, UVC and UVB upconversion emission peaked at 253, 273, 276, 279, 306 and 311 nm from ⁶D_J, ⁶I_J and ⁶P_J of Gd³⁺ can be obtained.

Figure 1 | Schematic energy diagram of heterogeneously doped lanthanide ions and their cascade energy transfer within a core-multishell nanoparticle. When the nanoparticles are excited under 808 nm, Nd³⁺ sensitizers first absorb the excitation energy and pass onto Yb³⁺. Subsequently, the ³P₂ state of Tm³⁺ is populated by a sequential five-photon energy transfer from the network of excited Yb³⁺ ions and relax to ¹I₆. The ⁶D_J state of Gd³⁺ is populated via a five-photon energy transfer process from Tm³⁺ and further populated by the energy transfer from Yb³⁺, giving rise to the sixth-photon upconversion luminescence. The inert NaYF₄ layer can lock-in the Gd³⁺ excitation energy and then reuse the energy that would otherwise be depopulated by deleterious energy traps within the nanoparticles, resulting in upconversion emission in middle UV region.

The interior energy flux leakage pathway through lattice vibration and multiphonon transitions can be neglected because of intrinsic low phonon energy of host materials (~350 cm⁻¹), and large energy gap of about 32,000 cm⁻¹ of Gd³⁺ compared with the intrinsic low phonon energy of host materials (~350 cm⁻¹) [Blasse, G.; Grabmaier, B. C. *Luminescent Materials*; Springer: Berlin, 1994].

Since the Gd³⁺-Gd³⁺ energy migration is efficient and it can travel long distances [*Nat. Mater.* **2011**, 10, 968], it is reasonable to assume that the excitation energy may be quenched by the interior lattice defects in the heterogenous structure with multi layers of shell. In our design, NaYF₄ in the first layer effectively block the energy transfer from Gd³⁺ to interior lattice defects in the outer shell layers, resulting in the migrating energy recycling in the core domain of NaGdF₄:Yb,Tm (Figure 6b). To validate our hypothesis, we investigated the lifetimes of Gd-CS_YS₂S₃, Gd-CS₁S_YS₃ and Gd-CS₁S₂S_Y

nanoparticles by changing the position the NaYF₄ layer. As shown in Figure 6c, the lifetime (Gd³⁺: 311 nm) in Gd-CS_YS₂S₃ is significantly longer than those in both Gd-CS₁S_YS₃ and Gd-CS₁S₂S_Y nanoparticles. The prolonged lifetime of Gd³⁺ in Gd-CS_YS₂S₃ nanoparticles is ascribed to the suppressed trapping of Gd³⁺ energy by interior lattice defects in the multi-shell regions. By contrast, a similar level of short lifetimes of Gd³⁺ in Gd-CS_{Gd}S₂S₃, Gd-CS₁S_YS₃, and Gd-CS₁S₂S_Y was observed, indicating that surface quenching was not responsible for the weak UV upconversion in conventional nanoparticles (Figure 6d). This result is also consistent with our luminescence analysis in that a significantly stronger UV luminescence of Gd-CS_YS₂S₃ nanoparticles compared with those of Gd-CS_{Gd}S₂S₃, Gd-CS₁S_YS₃ and Gd-CS₁S₂S_Y counterparts.

Furthermore, a Gd³⁺ content of 50 mol% produced an optimum energy-migration property with a Gd³⁺-Gd³⁺ separation of 5.32 Å, which can be approximately calculated using the following equation [*Nat. Photonics* **2020**, *14*, 760]:

$$d = \left(\frac{a^2 c \sqrt{3}/2}{1.5x} \right)^{1/3} \quad (3)$$

For the hexagonal NaGdF₄ unit cell, $a = 6.02 \text{ \AA}$, $c = 3.60 \text{ \AA}$. The short distance between Gd³⁺ ions indicates that the Gd³⁺-Gd³⁺ energy migration is dominated by exchange interaction [*Mater. Chem. Phys.* **1987**, *16*, 201]. Besides, it was reported by Blasse group, P_{em} for Gd³⁺ is about $5 \times 10^2 \text{ s}^{-1}$, $P_{(Gd^{3+} \rightarrow Gd^{3+})}$ is about 10^7 s^{-1} . P_{em} denotes the probability of emission, while $P_{(Gd^{3+} \rightarrow Gd^{3+})}$ denotes the probability for energy migration [*Mater. Chem. Phys.* **1987**, *16*, 201]. These results indicated that the excitation energy might be transferred more than 10^5 times for the excited Gd³⁺. After NaGdF₄ was replaced with NaYF₄ in the first layer, a significant increase in Gd³⁺ lifetime was observed (Figure 6c), suggesting the probability Gd³⁺-Gd³⁺ of energy migration within the core domain was significantly increased. Taken these together, the results conclusively suggested that the NaYF₄ shell can impede the fast migrating energy within the network of Gd³⁺ ions from being trapped by the interior lattice defects in the outer multi-layer shell, which promotes the occurrence of energy hopping in Gd³⁺ sublattice at the core domain of NaGdF₄:Yb,Tm, thereby realizing the intense UVB and UVC upconversion emissions. To achieve a more comprehensive understanding of upconverted excitation lock-in mechanism, we have added a detailed discussion of quantitative description to the revised manuscript and Supplementary Information.

Figure 6 | Determination of the interior traps and mechanistic investigation of Gd^{3+} energy recycling in heterogeneous core-shell nanoparticle. (a) Schematic illustration of the as-synthesized $\text{Gd-CS}_{\text{Gd}}\text{S}_{0\%}\text{NdS}_3$ nanoparticles, and the upconversion luminescence decay curves of Gd^{3+} emission at 311 nm and Tm^{3+} emission at 450 nm of $\text{Gd-CS}_{\text{Gd}}\text{S}_{50\%}\text{NdS}_3$ and $\text{Gd-CS}_{\text{Gd}}\text{S}_{0\%}\text{NdS}_3$ nanoparticles under 980 nm excitation, respectively. (b) Schematic illustration of energy recycling in Gd^{3+} sublattice at the core domain of $\text{NaGdF}_4:\text{Yb,Tm}$. (c) The upconversion luminescence decay curves of Gd^{3+} emission at 311 nm in $\text{Gd-CS}_Y\text{S}_2\text{S}_3$, $\text{Gd-CS}_1\text{Y}_1\text{S}_3$, and $\text{Gd-CS}_1\text{S}_2\text{S}_Y$ nanoparticles by pulsed 808 nm excitation, respectively. (d) The upconversion luminescence decay curves of Gd^{3+} emission at 311 nm in $\text{Gd-CS}_{\text{Gd}}\text{S}_2\text{S}_3$, $\text{Gd-CS}_1\text{SY}_1\text{S}_3$, and $\text{Gd-CS}_1\text{S}_2\text{S}_Y$ nanoparticles by pulsed 808 nm excitation, respectively.

Other aspects requiring an improvement:

Comment #3: 1) What are the solvent and the volume fraction in the solution of the NPs shown in Fig. 5? Is the efficiency of radiation-to-heat conversion similar to the two kinds of particles? (note that light is strictly used for radiation emitted in the visible part of the electromagnetic spectrum).

Response: We would like to thank the referee for pointing out this issue. In the original manuscript, the solvent was cyclohexane, and we kept the number concentrations of the two types of nanoparticles the same. As inspired by this referee, we realize that although the energy trapped at the defect sites will release heat via nonradiative decay, our original experiment design to study

thermal loading effects was unconvincing. Therefore, we have re-designed the experiment and identified the energy traps by the time decay study, as shown in Figure 6. To measure the efficiency of radiation-to-heat conversion of the two kinds of nanoparticles (Figure R4), we carefully conducted the experiment based on the previous reports (for referee only).

According to Prof. Roper's report [*J. Phys. Chem. C* **2007**, *111*, 3636], the photothermal conversion efficiencies η of Gd-CS_YS₂S₃ and Gd-CS_{Gd}S₂S₃ nanoparticles were calculated using the following equation:

$$\eta = \frac{hs(T_{max}-T_{amb})-Q_B}{I(1-10^{-A^{808}})}$$

(4)

Where h is heat transfer coefficient, S is the surface area of the container, T_{max} is the equilibrium temperature, T_{amb} is ambient temperature of the surroundings. Q_B is heat lost from light absorbed by the container itself, which was measured independently containing pure cyclohexane without UCNPs of Gd-CS_YS₂S₃ and Gd-CS_{Gd}S₂S₃. And A^{808} is the absorption intensity of Gd-CS_YS₂S₃ and Gd-CS_{Gd}S₂S₃ at 808 nm. The value of hS is derived according to the following equation:

$$\tau_s = \frac{m_H C_H}{hs} \quad (5)$$

Where τ_s is the sample system time constant, m_H and C_H are the mass and heat capacity of cyclohexane used as the solvent, respectively. And, τ_s can be calculated by the following equation:

$$t = -\tau_s \ln \theta \quad (6)$$

Here θ is defined as the expression below:

$$\theta = \frac{T-T_{amb}}{T_{max}-T_{amb}} \quad (7)$$

Time constant of Gd-CS_YS₂S₃ and Gd-CS_{Gd}S₂S₃ for heat transfer from the system is respectively determined to be $\tau_s = 167$ s and 188 s applying the linear time data from the cooling period (after 300 s) vs negative natural logarithm of driving force temperature (Figure R4b and R4d). Substituting the value of τ_s into eq 2, hS can be obtained. And the value of hS replaced into eq 1, 808 nm laser heat conversion efficiencies (η) of Gd-CS_YS₂S₃ and Gd-CS_{Gd}S₂S₃ nanoparticles can be respectively calculated to be 7.29% and 6.33%.

Figure R4. Efficiency of radiation-to-heat conversion analysis. (a) Photothermal effect of a cyclohexane of Gd-CS_{Ga}S₂S₃ nanoparticles (concentration: 40 mg/ mL) when illuminated with an 808-nm laser (10 W/cm²). The laser was turned off after irradiation for 5 min. (b) Plot of cooling time versus negative natural logarithm of the temperature difference, obtained from the cooling stage as shown in (a). The time constant for heat transfer of the system is determined to be $\tau_s = 188$ s. (c) Photothermal effect of a cyclohexane of Gd-CS_yS₂S₃ nanoparticles (concentration: 40 mg/ mL) when illuminated with an 808-nm laser (10 W/cm²). The laser was turned off after irradiation for 5 min. (d) Plot of cooling time versus negative natural logarithm of the temperature difference, obtained from the cooling stage as shown in (c). The time constant for heat transfer of the system is determined to be $\tau_s = 167$ s.

Comment #4: 2) What is the meaning of the full lines in Fig. S1? What function is used to fit the size distributions?

Response: We thank the reviewer for the comments. Nanoparticle size distribution is fitted by a Gaussian curve. The full lines are Gaussian curve. The function of Gaussian curve is to calculate average nanoparticles size.

Comment #5: 3) It is difficult to grasp the message from Figs. S9 and S10b (all the bars have the same height...)

Response: We thank the reviewer for the comments. As suggested, we have revised them as shown in the manuscript accordingly (Supplementary Figures 9 and 10b).

Figure R5. The comparison upconversion emission intensity. The enhancement factor of the emission at 290 and 311 nm obtained by comparing the results for Gd-CS_YS₂S₃ and Gd-CS_{Gd}S₂S₃ nanoparticles under excitation of 808 nm (a) and (b) CW diode laser. The excitation power density is 10 W cm⁻².

Comment #6: 4) What are the experimental conditions of the absorption spectra shown in Fig. S10 (solvent, NPs volume fraction, reference, etc)? To support the sentence “(...) although the absorption profile of Gd-CS_YS₂S₃ is not changed compared with that of Gd-CS_{Gd}S₂S₃ nanoparticles (Supplementary Fig. S11)” and, thus, validate the qualitative comparison of intensities between different samples authors must record their absorbance in absolute units (not arb. units). Otherwise, the measurement of emission absolute quantum yields is mandatory.

Response: We thank this reviewer for the careful reading and valuable comments on our manuscript. To avoid the uncertainty in the process of ligand removal, we directly measured absorption spectra of nanoparticles in cyclohexane. To compare the absorption profile of Gd-CS_YS₂S₃ and Gd-CS_{Gd}S₂S₃ nanoparticles, we have double confirmed this data by recording the absorption spectra of the two kind of nanoparticles under an identical number concentration of these nanoparticles. As shown in Supplementary Figure 11, the absorption profile of Gd-CS_YS₂S₃ is not changed compared with that of Gd-CS_{Gd}S₂S₃ nanoparticles. The approximate absorption cross-section σ of Nd³⁺ at 808 nm was calculated from the UV-vis absorption spectra of the nanoparticles using the following equations [*J. Cryst. Growth* **2003**, 252, 241]:

$$A = \varepsilon \cdot M \cdot l \quad (7)$$

$$\sigma = \frac{\varepsilon}{n} \quad (8)$$

Where A is absorbance, ε is the absorption coefficient, M is the molar concentration of Nd³⁺, l is the path length, n is atomic number density of Nd³⁺ ions. $\sigma = 1.5 \times 10^{-19}$ cm² (808 nm for Gd-CS_YS₂S₃), $\sigma = 1.3 \times 10^{-19}$ cm² (808 nm for Gd-CS_{Gd}S₂S₃). These results suggested that the absorption cross-section of Nd³⁺ in Gd-CS_YS₂S₃ were essentially unchanged compared with that of in Gd-CS_{Gd}S₂S₃.

Moreover, although hexagonal phase NaYF₄ and NaGdF₄ adopt the same space group (P63/m), the unit-cell parameters of NaYF₄ and NaGdF₄ are slightly different (a = 5.96 Å, c = 3.53 Å for NaYF₄ and a = 6.02 Å, c = 3.60 Å for NaGdF₄) [*Nature*, **2010**, 463, 1061], thus we didn't make the volume fraction the same to measure their absorption spectra.

Furthermore, we have conducted the quantitative study for Gd-CS_YSS and Gd-CS_{Gd}SS nanoparticle to compare their quantum yield. We measured the quantum yield on a FLS1000 equipping with integrating sphere (Edinburgh), in conjunction with 808 nm diode lasers and an integrating sphere via an absolute method. The estimated upconversion quantum yields from 240 to 750 nm of the as-prepared Gd-CS_YS₂S₃ and Gd-CS_{Gd}S₂S₃ nanoparticles were approximately 1.74% and 0.97%, respectively. To quantitatively study the emission enhancement in the UV range from 240 to 325 nm, we also tried to measure the upconversion quantum yields in the UV range. However, these nanoparticles showed a weak UV upconversion emission, leading to undetectable quantum yields. Instead, we measured the quantum yields of upconversion emission in the range of 240-400 nm of the as-prepared Gd-CS_YS₂S₃ and Gd-CS_{Gd}S₂S₃ nanoparticles, and the results were approximately 0.13% and 0.04%, respectively.

Supplementary Figure 11. Comparative absorption spectroscopic studies of Gd-CS_YS₂S₃ and Gd-CS_{Gd}S₂S₃ nanoparticles. Absorption spectrum of Gd-CS_YS₂S₃ and Gd-CS_{Gd}S₂S₃ nanoparticles in cyclohexane with the same number concentration.

Comment #7: 5) A short paragraph reviewing the work performed in Nd³⁺-sensitized UCNPs must be added.

Response: As suggested, we have added a short discussion to our revised manuscript. Compared with Yb³⁺-sensitized upconversion nanoparticles (UCNPs), Nd³⁺-sensitized UCNPs offer deep penetration depths and minimal over-heating effect, owing to low coefficients of water absorption under 800-nm excitation [*ACS Nano* **2013**, 7, 7200]. Nd³⁺-sensitized UCNPs are the promising candidates for photon-driven reactions in various biosystems, such as biodetection [*ACS Appl. Mater. Interfaces* **2019**, 11, 7441;], photodynamic therapy [*ACS Nano* **2017**, 11, 4133; *ACS Nano* **2018**, 12, 3217; *Chem. Sci.* **2018**, 9, 3141; *Angew. Chem. Int. Ed.* **2020**, 59, 2634], light-triggered drug release [*ACS Nano* **2017**, 11, 2846], and photocatalysis [*J. Phys. Chem. C* **2020**, 124, 18081]. To enhance the brightness of Nd³⁺-sensitized UCNPs, core-shell nanostructural design has been typically utilized to prevent deleterious cross-relaxation [*J. Am. Chem. Soc.* **2013**, 135, 12608; *Angew. Chem. Int. Ed.* **2013**, 52, 13419; *Adv. Opt. Mater.* **2013**, 1, 644; *Adv. Mater.* **2014**, 26, 2831]. By doping lanthanide ions and Nd³⁺ ions into

separated layer, the emission intensity can be greatly enhanced while maintaining optical integrity [*Nat. Commun.* **2018**, *9*, 2415]. Despite enticing prospects, UVC emission from Nd³⁺-sensitized UCNPs has been challenging because of the densely packed excited states of Nd³⁺ and dominant cross-relaxation within the nanoscale systems [*J. Phys. Chem. Lett.* **2020**, *11*, 2883]

Reviewer #2 (Remarks to the Author):

Comment #1: This is an interesting paper in which authors have designed and synthesized a new type of core-shell structure capable of intense UV emission (below 300 nm). For that they have used a complex structure in which the presence of Gd ions serve as energy reservoir. The idea itself is interesting. The materials prepared are of outstanding quality. The text is well-written and the figures have been nicely prepared and designed.

The final result is a nanoparticle capable of UV emission under 808 nm radiation. At the beginning of their manuscript authors stated "Deep-UV upconversion emission is particularly useful for drug release in deep tissues..." I think that the paper needs such demonstration. I recognize the impact of the paper and its relevance. But it is not demonstrated that the resulting nanostructures can have a practical application. In particular, it would be nice to demonstrate the application of these structures where other have failed. I think the impact and quality of the paper could be improved by orders of magnitude by including some evidence of the advances that can be achieved by such structures.

Response: We thank this reviewer for the positive comments. This study was mainly focused on gaining a fundamental understanding the optical process of the designed heterogeneous core-shell structures leading to the enhanced UV generation. The potential applications of the nanoparticles with intense UV emission may include ultraviolet solid-state lasers, photocatalysis [*J. Am. Chem. Soc.* **2019**, *141*, 7056; *Nat. Commun.* **2016**, *7*, 10304; *Appl. Catal. B* **2019**, *243*, 438] and nanomedicine [*Acc. Chem. Res.* **2014**, *47*, 3052; *J. Am. Chem. Soc.* **2013**, *135*, 18920; *Adv. Mater.* **2016**, *28*, 9341], which is out of the scope of this work and our current expertise. To satisfy this reviewer's suggestion, in our revised manuscript, we prepared the titanium dioxide (TiO₂)-coated UCNPs in which TiO₂ serve as the photosensitizer (*ACS Nano* **2015**, *9*, 2584). TEM and X-ray powder diffraction (XRD) analysis confirmed the successful synthesis of TiO₂-coated upconversion nanoparticles (Supplementary Figure 24a,b). Compositional analysis of these nanocomposites by energy-dispersive X-ray spectroscopy (EDX) confirms the presence of Ti⁴⁺, Nd³⁺, Gd³⁺, Yb³⁺, and Tm³⁺ (Supplementary Figure 25). The ability to generate singlet oxygen (¹O₂) of the as-synthesized nanocomposites was evaluated by the 1,3-diphenylisobenzofuran (DPBF) chemical probe under 808 nm laser irradiation. As shown in Supplementary Fig. 26a, the emission Gd-CS₂S₃@TiO₂ become weaker compared with Gd-CS₂S₃ nanoparticles due to the absorbance of UV emission by TiO₂ shell. As shown in Supplementary Figure 26b, the characteristic absorption decreased with time, indicating the generation of ¹O₂. These results indicate the enticing prospects of NIR light mediated photosensitizing nanocomposites for many ROS mediated bio-applications.

Supplementary Figure 24. (a) TEM images of the as-synthesized Gd-CS_yS₂S₃@TiO₂ nanocomposites. (b) XRD pattern of the as-synthesized Gd-CS_yS₂S₃@TiO₂ nanocomposites.

Supplementary Figure 25. Energy dispersive X-ray (EDX) spectrum of Gd-CS_yS₂S₃@TiO₂ nanocomposites.

Supplementary Figure 26. (a) Emission spectra of Gd-CS_YS₂S₃, Gd-CS_YS₂S₃@TiO₂ under 808 nm excitation. (b) Absorbance changes of DPBF treated with Gd-CS_YS₂S₃@TiO₂ nanocomposites accompanied with 808 nm laser irradiation for different times.

Comment #2: In connection with the previous comment, authors are claiming that their design improves the final UV emission intensity. They compare in the main text the emission spectra in presence and absence of Gd showing the UV emission improvement. But from that figure it is difficult to have a quantitative idea of how big is the improvement. I think the paper will benefit from a precise qualification of the improvement achieved. Also if a number is given in main text about QY, some comparison with previous results are required.

Response: We thank the reviewer for this constructive and helpful comments. We have conducted a quantitative study compare the quantum yields of Gd-CS_YSS and Gd-CS_{Gd}SS nanoparticle. We measured the quantum yield on a FLS1000 equipping with integrating sphere (Edinburgh), in conjunction with 808 nm diode lasers and an integrating sphere via an absolute method. The upconversion quantum yields from 240 to 750 nm of the as-prepared Gd-CS_YS₂S₃ and Gd-CS_{Gd}S₂S₃ nanoparticles were estimated as 1.74% and 0.97%, respectively. To quantify the emission enhancement in the UV range from 240 nm to 325 nm, we also attempted to measure the upconversion quantum yields in the UV range, but, without success due to the limited UVC emissions. Instead, we measured the quantum yields of upconversion emissions in the range from 240 nm to 400 nm with the results being approximately 0.13% and 0.04%, respectively. Limited by the sensitivity of the instrument, the above quantitative numbers of absolute QYs are at least consistent to our previous spectroscopy observations of the relative intensities.

Comment #3: Finally, the discussion about thermal loading effects is confusing. All along the work authors state that they use 808 nm (Nd) absorption to avoid the heating of 980 nm radiation (Yb) but then data included in the figure 5 seems to indicate the opposite. In general this figure and related discussion should be re-designed to make it clear. Also, the analysis of the heating as a function of pump power should be included to evaluate the different heat pathways that are different from the one included in Figure 5

In summary, it is a good piece of work and I think the authors can make it even better and suitable for Nature Comm if they include some new experiments to show the practical potential of the structure as well as to quantify the improvement they have achieved.

Response: We would like to thank the referee for the supportive comments on our work. To clarify the confusion associated with Figure 5, we redesigned the thermal loading effect experiment. We utilized different power density of 808 nm and 980 nm to excite Gd-CS₂S₂S₃ and DI water to measure the heating effect of these solutions. As shown in Figure R6, the temperature rises of water determined to be 0.7 and 9.9 °C under continuous irradiation with 808 and 980 nm lasers at 10 W/cm², respectively. Similarly, the temperature rises of the aqueous solution with nanoparticles were measured to be 2.6 and 11.4 °C under identical irradiation condition with 808 and 980 nm lasers, respectively. In addition, the same temperature rises trends were observed by using different excitation power density (for referee only).

Figure R6. Heating effect of nanoparticles in water and aqueous solution under 808 nm and 980 nm laser irradiations. Temperatures of water and the as-prepared nanoparticles Gd-CS₂S₂S₃ in aqueous solution under continuous irradiation with 808 or 980 nm laser for 5 min, the laser power density is 10 W/cm² (a-b), 8.6 W/cm² (c-d), and 7.6 W/cm² (e-f), respectively.

Reviewer #3 (Remarks to the Author):

Comment #1: Most upconversion systems developed so far concentrate in converting near-infrared light into visible and near UV spectral ranges. However, light with shorter wavelength, in the UV-C range has many industrial and medical applications. Here the authors propose a sophisticated type of core-multishell upconverting nanoparticles which could potentially be provided for in situ production of singlet oxygen and/or drug release in photodynamic treatment of cancer.

This described work is an expansion of a study published last year in *Nanoscale* (ref. 30) in which Gd-C_(YbTm)S_(Yb)S_(YbNd)S_(Gd) nanoparticles, based on the NaGdF₄ host, were described and showed to have more intense Tm emission under 808-nm (Nd) excitation compared with 980-nm (Yb) excitation; additionally transformation in light onto heat was also demonstrated. In the present work, the authors replace the host material of the first shell, NaGdF₄ with optically inactive NaYF₄, while keeping all dopant concentrations the same. This simple modification has a dramatic effect on the photophysical properties in that now Gd ions can be excited by a 6-photon process.

The experimental section is convincing and all necessary control experiments have been performed, so that the result is well proved. The paper demonstrates that seemingly minor, but well thought, modifications of hetero nanostructures can lead to substantial beneficial modifications of their photophysical properties and opens large perspectives in the field.

Response: We would like to thank this reviewer for careful reading and valuable comments on our manuscript.

Comment #2: I am missing some more quantitative aspects, regarding efficiencies of energy transfer and overall quantum yield for instance; uncertainties on reported data are also missing as well as the number of repeat experiments made.

Response: We thank the reviewer for the constructive comments. We have seriously taken this reviewer's comments and conducted the quantitative studies on the efficiencies of all the energy transfer processes. The large energy gap of about 32000 cm⁻¹ of Gd³⁺ and intrinsic low phonon energy of NaGdF₄ offer good possibilities to obtain 100% energy transfer efficiency from Gd³⁺-to-Gd³⁺ [*Mater. Chem. Phys.* **1987**, *16*, 201; Blasse, G.; Grabmaier, B. C. *Luminescent Materials*; Springer: Berlin, 1994. P100]. The energy transfer efficiencies η of Nd³⁺-to-Yb³⁺, Yb³⁺-to-Tm³⁺, and Tm³⁺-to-Gd³⁺ have been primarily estimated from eq 1 and eq 2 [*Phys. Rev. B* **1971**, *4*, 3153; *J. Lumin.* 2015, 166, 177]:

$$\eta = 1 - \frac{\tau_m}{\tau_{Ln}} \quad (1)$$

$$\tau_m = \frac{\sum \alpha_i \tau_i^2}{\sum \alpha_i \tau_i} \quad (2)$$

Where τ_m is the mean lifetime of energy donor lanthanides (Ln) in the presence of energy acceptor, τ_{Ln} is the intrinsic lifetime of energy donor, and α is the amplitude. To calculate the energy transfer efficiencies of Nd³⁺-to-Yb³⁺, Yb³⁺-to-Tm³⁺, and Tm³⁺-to-Gd³⁺, we designed and synthesized three pairs of heterogeneous nanoparticles (TEM results shown in Supplementary Fig. 14). In our experiment, to first determine the intrinsic lifetime of the corresponding energy donors, the energy acceptors of Yb³⁺, Tm³⁺, and Gd³⁺ were replaced by optically inert Y³⁺ ions.

In details, to calculate the energy transfer efficiency of Nd³⁺-to-Yb³⁺, we produced a pair of samples of 49%Yb,50%Gd,1%Tm@80%Y,20%Yb@10%Yb,50%Nd,40%Gd@100%Gd (Gd-CS₇SS in the presence of 20% Yb³⁺ energy acceptor) v.s. 49%Y,50%Gd,1%Tm@100%Y@10%Y,50%Nd,40%Gd@100%Gd (Gd-

CS_YSS in the absence of 20% Yb³⁺). The lifetimes of Nd³⁺ at 893 nm were measured under the 793 nm pulsed excitation, and the energy transfer efficiency of Nd³⁺-to-Yb³⁺ was calculated to be 79% (Figure 3a). Similarly, to calculate the energy transfer efficiency of Yb³⁺-to-Tm³⁺, and to avoid the complex energy transfer pathways in the core-multishell structure (Figure R2), we produced a pair of simplified designs of 50%Gd,20%Yb,1%Tm,29%Y@100%Y (Gd-CS_Y in the presence of 1% Tm³⁺ energy acceptor) v.s. 50%Gd,20%Yb,30%Y@100%Y (Gd-CS_Y in the absence of Tm³⁺). The 980 nm decay lifetimes of Yb³⁺ were measured under the 920 nm pulsed excitation, and the energy transfer efficiency of Yb³⁺-to-Tm³⁺ was estimated to be 62% (Figure 3b). To calculate the energy transfer efficiency of Tm³⁺-to-Gd³⁺, we produced a pair of samples of 50%Gd,20%Yb,1%Tm,29%Y@100%Y (Gd-CS_Y in the presence of Gd³⁺) v.s. 79%Y,20%Yb,1%Tm@100%Y (Gd-CS_Y in the absence of Gd³⁺). By exciting the samples at 980 nm, the lifetimes of Tm³⁺ at 290 nm were and the energy transfer efficiency of Tm³⁺-to-Gd³⁺ was estimated to be 1% (Figure 3c).

Figure 3 | Energy transfer efficiency of Nd³⁺-to-Yb³⁺, Yb³⁺-to-Tm³⁺ and Tm³⁺-to-Gd³⁺ ion. (a) Luminescence decay curves of Nd³⁺ emissions measured at 893 nm for 49%Yb,50%Gd,1%Tm@80%Y,20%Yb@10%Yb,50%Nd,40%Gd@100%Gd (with Yb³⁺) and 49%Y,50%Gd,1%Tm@100%Y@10%Y,50%Nd,40%Gd@100%Gd (without Yb³⁺) by pulsed 793 nm excitation. (b) Luminescence decay curves of Yb³⁺ emissions measured at 980 nm for 50%Gd,20%Yb,1%Tm,29%Y@100%Y (with Tm³⁺) and 50%Gd,20%Yb,30%Y@100%Y (without Tm³⁺) by pulsed 920 nm excitation. (c) Luminescence decay curves of Tm³⁺ emissions measured at 290 nm for 50%Gd,20%Yb,1%Tm,29%Y@100%Y (with Gd³⁺) and 79%Y,20%Yb,1%Tm@100%Y (without Gd³⁺) by pulsed 980 nm excitation.

We have further conducted the quantitative study for Gd-CS_YSS and Gd-CS_{Gd}SS nanoparticle to compare their quantum yields. Quantum yield measurements were conducted on a FLS1000 equipping with integrating sphere (Edinburgh), in conjunction with 808 nm diode lasers and an integrating sphere via an absolute method. The estimated upconversion quantum yields from 240 to 750 nm of the as-prepared Gd-CS_YS₂S₃ and Gd-CS_{Gd}S₂S₃ nanoparticles were approximately 1.74% and 0.97%, respectively. To quantitatively study the emission enhancement in the UV range from 240 to 325 nm, we also attempted to measure the upconversion quantum yields in the UV range. However, these nanoparticles showed a weak UV upconversion emission, leading to undetectable quantum yields. Instead, we measured the quantum yields of upconversion emission in the range of 240-400

nm of the as-prepared Gd-CS₇S₂S₃ and Gd-CS_{Gd}S₂S₃ nanoparticles, and the results were approximately 0.13% and 0.04%, respectively.

Key experiments were repeated more than three times and all the other experiments were repeated twice. The synthesis of Gd-CS₇S₂S₃ and Gd-CS_{Gd}S₂S₃ and their optical studies have been repeated by three different students, suggesting the results are repeatable. The uncertainties and the numbers of repeat experiments have been added and discussed in the revised manuscript.

Comment #3: Remarks a. I question the use of “deep UV” for the spectral range of the upconverted wavelength produced by the described core-multi-shell nanoparticles. This denomination is too vague. In classical nomenclature, the UV range (400-10 nm) is subdivided between near UV (400-300 nm), middle UV (300-200 nm), and far UV (200-10 nm; sometimes subdivided in far UV – 200-100 nm and extreme UV – 100-10 nm). Therefore since the upconverted light described in this paper has wavelength around 250 nm, “deep UV” should be replaced with “middle UV”. Alternatively, the authors could use the other classification, namely UV-C (100-290 nm).

Response: As suggested, we have replaced “deep UV” with “UV-C” in the revised manuscript accordingly.

Comment #4: b. I am puzzled by the use of “optically inactive shell” or “optical inert interlayer” for the first NaYF₄ shell because its composition is NaYF₄:Yb(20%) and it acts as an energy migrator relay, so the shell is not optically inactive, but the host matrix is. I do not quite understand. Is the first shell homogeneous, i.e. entirely comprised of NaYF₄:Yb(20%) or is it heterogeneous with a first undoped layer? Please clarify.

Response: Indeed, the host matrix NaYF₄ is optically inactive, while the first layer was doped with 20% Yb. To state more accurately, we have rephrased the sentences accordingly. See page 3, line 20 and line 22; page 4, line 7 and line 8, etc.

Comment #5: c. Figure 1, caption. Please replace “6-photon” with “6th photon” since the transfer from Tm to Gd is a one-photon excitation.

Response: We thank the referee for the comments. As suggested, we have revised it accordingly. See page 4, line 17; page 22, Figure 1, caption, line 6.

Comment #6: d. Results. First section. The description of the phenomena occurring in the nanoparticles lacks precision. For instance in the third sentence why describing energy transfer to Tm as being a back energy transfer? Back transfer from Yb would be transfer to Nd. Fourth sentence: Why write “By doping of Gd” since Gd is in the host material?

Response: We would like to thank this reviewer for raising the concerns. We have rephrased the sentences accordingly. See page 3, Results, line 5 and line 7.

Comment #7: e. Results, second section. It would be clearer for the reader to give the composition of the nanoparticles used in this work rather than the one of those described in ref. 30, as is done in the “controlled synthesis” section.

Response: As suggested, we have revised it accordingly. See page 3, results, line 10.

Comment #8: f. Figure 2d. I am surprised by the semi-log plot. Usually a log-log plot should be used to determine the number of implied photons, as one can easily imply from the formula given on page 6 (2nd line).

Response: We thank the reviewer for the constructive comments. We have re-conducted the experiment and replaced the original figure with a log-log plot (Figure 2g and Supplementary Figure 13).

Figure 2g. Determination of the number of implied photons. Log intensity-pump power of the 253 nm upconversion emission of Gd-CyS₂S₃ nanoparticles.

Supplementary Figure 13. (a,b) Log intensity-pump power of the 276 and 310 nm upconversion emission of Gd-CyS₂S₃ nanoparticles under 808 nm excitation, respectively.

REVIEWERS' COMMENTS

Reviewer #1 (Remarks to the Author):

The authors performed a detailed revision of the initial submission performing new experiments and including new results that substantially improved the manuscript. Moreover, the majority of the referees' concerns were adequately addressed. In short, the manuscript is a nice piece of work and I recommend its publication.

Reviewer #2 (Remarks to the Author):

I am happy with this revision. I appreciate the inclusion of some results on photocatalytic activity based on these novel structures.

I think that the paper is ready for publication as authors have considered all my previous comments.

Reviewer #3 (Remarks to the Author):

Following the reviewers' remarks, the authors have substantially upgraded their manuscript, in particular by adding new experiments and data on energy transfer processes and their mechanisms, quantum yields, reactive oxygen species generation, and thermal loading effects. The result is a much improved manuscript which, in my opinion is ready for publication in Nat. Commun.